# An Enhanced SPEI Drought Monitoring Method Integrating Land Surface Characteristics

Liqing Peng[1,2,3*], Justin Sheffield[4], Zhongwang Wei[5], Michael Ek[6], Eric F. Wood[7†]

[1] Food Program, World Resources Institute, Washington DC, United States.
[2] Department of Geography, The University of Hong Kong, Hong Kong, China.
[3] Institute for Climate and Carbon Neutrality, The University of Hong Kong, Hong Kong, China.
[4] School of Geography and Environmental Science, University of Southampton, Southampton, United Kingdom.
[5] Southern Marine Science and Engineering Guangdong Laboratory (Zhuhai), Guangdong Province Key Laboratory for Climate Change and Natural Disaster Studies, School of Atmospheric Sciences, Sun Yat-Sen University, Guangzhou, China.
[6] Joint Numerical Testbed, Research Applications Laboratory, National Center for Atmospheric Research, Boulder, Colorado, United States.
[7] Department of Civil and Environmental Engineering, Princeton University, Princeton, New Jersey, United States.
(†Deceased)

*Correspondence to*: Liqing Peng (pengliqing51@gmail.com)

**Abstract.** Atmospheric evaporative demand is a key metric for monitoring agricultural drought. The existing ways of estimating evaporative demand in drought indices do not faithfully represent the constraints of land surface characteristics and become less accurate over non-uniform land surfaces. This study proposes incorporating surface vegetation characteristics, such as vegetation dynamics data, aerodynamic and physiological parameters, into existing potential evapotranspiration (PET) methods. This approach is implemented over the Continental United States (CONUS) for the period of 1981-2017 and is tested in a recently developed drought index the Standardized Precipitation Evapotranspiration Index (SPEI). We show that activating realistic maximum surface and aerodynamic conductance could improve prediction of soil moisture dynamics and drought impacts by 29-41% on average compared to the widely used simple methods, especially effective in the forests and humid regions, by 86-89%. Surface characteristics that have a strong influence on the performance of the SPEI are mainly driven by leaf area index (LAI). Our approach only requires the minimum amount of ancillary data, while permitting both historical reconstruction and real-time forecast of drought. This offers a physically meaningful, yet easy-to-implement way to account for the vegetation control in drought indices.

## 1 Introduction

Drought is one of the most costly hydrological hazards (Wilhite, 2000; Ross & Lott, 2003; Piao et al., 2019), with devastating impacts on croplands and pastures (Kogan, 1995), forests ecosystems (Clark et al., 2016; Xu et al., 2022), electricity production, water quality, and soil fertility (Loon, 2015). Monitoring the changes in water availability is critical for providing early warnings of drought and for risk management (Wilhite, Sivakumar, & Pulwarty, 2014). Many physical or probabilistic measures have been developed (Heim, 2002) to quantify drought, such as Palmer Drought Severity Index (PDSI, Palmer, 1965), Standardized Precipitation Index (SPI, McKee, Doesken, Kleist, & others, 1993), Vegetation Condition Index (VCI, Kogan, 1995), and multiple remote sensing drought indices (Zhang, Jiao, Zhang, Huang, & Tong, 2017; Yang et al., 2023).

Atmospheric evaporative demand (AED) is a key input to drought indices because it is a measure of water demand, namely, how thirsty the atmosphere is (Peng, Li, & Sheffield, 2018). AED typically reflects the effect of temperature and humidity, and is considered a major driver of drought stress on vegetation and tree mortality (Williams et al., 2012; McDowell et al., 2018). Among the drought indices, the recently developed Standardized Precipitation Evapotranspiration Index (SPEI) (Vicente-Serrano, Beguería, & López-Moreno, 2010) factors in water demand (AED) in addition to water supply (precipitation). Compared to the SPI that only considers precipitation, the SPEI is more suitable for quantifying the drought impacts on agriculture (Potop, 2011; Potop, Možný, & Soukup, 2012), and ecosystems (Vicente-Serrano et al., 2012; Vicente-Serrano et al., 2013; Barbeta, Ogaya, & Peñuelas, 2013). In addition, the SPEI is more flexible than the PDSI because it is not sensitive to soil water field capacity and can be implemented on various time scales (Vicente-Serrano, der Schrier, Beguería, Azorin-Molina, & Lopez-Moreno, 2015; Zhao et al., 2017). It has been widely used for both drought reconstruction and monitoring (Paulo, Rosa, & Pereira, 2012; Beguería, Vicente-Serrano, Reig, & Latorre, 2013).

The way of estimating AED in drought indices has a significant impact on drought quantification (Sheffield, Wood, & Roderick, 2012; Trenberth et al., 2013; Yang, Roderick, Zhang, McVicar, & Donohue, 2018; Dewes et al., 2017). AED is approximated by potential evapotranspiration (PET), the maximum rate of evapotranspiration when surface water supply is unlimited. Previous work has used various PET formulations for AED in the SPEI since it was first proposed in 2010 (Vicente-Serrano, Beguería, & López-Moreno, 2010; Beguería, Vicente-Serrano, Reig, & Latorre, 2013). These conventional PET methods do not factor in the effects of surface characteristics, which often assume no or simple universal vegetation control on transpiration (e.g., the Thornthwaite, Hargreaves-Samani, and Penman methods). Without vegetation control, the maximum surface conductance is overestimated and the PET rate during the onset and retreat of the

growing season is unrealistically high. Furthermore, by assuming an smooth reference surface, some methods do not account for surface roughness, hence downplay aerodynamic conductance and suppress the PET estimate (Peng et al., 2019). Even though the reference evapotranspiration ($ET_0$) method (Allen, Pereira,

Raes, & Smith, 1998) considers the biophysical limitation of transpiration by assigning a surface resistance under well-watered condition, it does not account for vegetation phenology (Lorenz, Davin, Lawrence, Stöckli, & Seneviratne, 2013) and assumes a fixed reference height and a constant surface resistance for all vegetation types. This approach is not physically meaningful for forests, where canopy height is relatively large and vegetation cover varies significantly. A recent study by Sun et al. (2023) highlighted the importance

of incorporating surface properties especially vegetation control in PET and used a two source Shuttleworth-Wallace (SW) model designed and validated for sparse and fragmented vegetation surfaces. However, without further calibration and parameterization, the SW model's broader applicability beyond sparse vegetation is uncertain, and additionally it may increase data requirements and associated uncertainties (Gao et al., 2021; Abeysiriwardana et al., 2022).

We hypothesize that adding the surface vegetation characteristics to an existing drought quantification approach will improve the spatial and temporal accuracy of drought prediction. The goals of this study are to explore which surface features are the most useful for enhancing drought prediction, and which vegetation types benefit most from incorporating these features. We propose incorporating realistic vegetation restrictions into existing PET methods, while not increasing much cost and uncertainty caused by additional

data sources and complex formulations. Then we use independent soil moisture observations (Dai, Trenberth, & Qian, 2004) from satellite to evaluate the drought depictions by various forms of PET approaches across different temporal scales. The evaluation against observed soil moisture allows the direct diagnosis of the most sensitive surface characteristics and the most effective approach for drought quantification (Vicente-Serrano et al., 2012).

In this study, we focus on the continental U.S. (CONUS) primarily because the drought events hitting this region have raised interest in variability, trends, and future risks of drought (Andreadis & Lettenmaier, 2006; Hobbins et al., 2012; Dewes et al., 2017). Several most severe droughts hit the western U.S. in the recent decade, including the 2012 Great Plains drought (Hoerling et al., 2014) and the 2012-2016 California drought (Dong et al., 2019). The western U.S. has been experiencing the most severe drought period after the 1930s

and 1950s (Andreadis, Clark, Wood, Hamlet, & Lettenmaier, 2005), and its vulnerability to drought continued to grow (Andreadis & Lettenmaier, 2006). Besides, high-quality meteorological datasets are available over the CONUS (Daly et al., 2008; Xia et al., 2012) and can help reduce the uncertainty of drought prediction originating from input forcings.

**2 Data**

 **2.1 Meteorology**

To calculate the SPEI, PET is estimated on daily scale over the period of 1981-2017 using high-quality daily meteorology data from PRISM (Parameter-elevation Regressions on Independent Slopes Model) that employs weather stations and digital elevation model (Daly, Neilson, & Phillips, 1994; Daly et al., 2008). We acquire daily precipitation, daily mean, maximum, minimum, and dew point temperature on a 4 km grid

for the period of 1981-2017. Surface downward shortwave and net longwave radiation, pressure, and wind speed are taken from the NLDAS-2 (North American Land Data Assimilation System phase 2 (Xia et al., 2012). All data are spatially restricted to the continental United States (25–50ºN, 67–125ºW) and regridded to the 0.125º NLDAS-2 grid using the first-order conservative remapping tool provided by Climate Data Operators (https://code.zmaw.de/projects/cdo).

**2.2 Soil moisture**

The European Space Agency Climate Change Initiative (ESA CCI) v4.3 surface soil moisture (SMsurf) is used to evaluate the drought severity quantified by the SPEI time series (https://www.esa-soilmoisture-cci.org/). This dataset combines several active and passive microwave soil moisture products into a harmonized surface layer soil moisture (2-5 cm) in $m^3 \, m^{-3}$ (Liu et al., 2012; Gruber et al., 2017). The dataset

is chosen for its enhanced data reliability by integrating multiple single-sensor active and passive microwave soil moisture products to minimize uncertainty (Gruber et al., 2019). The version 4.3 provides soil moisture on a 0.25º grid at daily time step for the 1979-2017 period and has been widely used in ET and drought studies (Dorigo et al., 2017; Martens et al., 2017).

**2.3 Land surface ancillary data**

The land surface data used for deriving biophysical parameters include gridded land cover type, leaf area index, and surface albedo. The land cover type is provided by the 0.5 km MODIS-based Global Land Cover Climatology during the 2001-2010 period (Broxton, Zeng, Sulla-Menashe, & Troch, 2014, https://archive.usgs.gov/archive/sites/landcover.usgs.gov/global_climatology.html). This dataset has 17 land cover classes based on the International Geosphere‐Biosphere Program (IGBP) classification. This land

cover climatology dataset is displayed in Fig. 1.

The monthly climatology of leaf area index is obtained from the 15-day, 1 km AVHRR GIMMS LAI3g product that covers the period of 1982-2016 (Zhu et al., 2013).

The monthly climatology of surface albedo is derived from the 8-day, 0.05º GLASS (Global Land Surface Satellite) albedo product. This GLASS02A05/06 product combines MODIS and AVHRR (Advanced Very-High-Resolution Radiometer) to provide a gap-filled land surface shortwave black-sky and white-sky albedo (Qu et al., 2014; Liu et al., 2013) that covers the period of 1982-2012. We resample the 8-day albedo to a daily resolution and obtain daily albedo by averaging the black- and white-sky albedos. Missing data are gap-filled using the average of adjacent years.

This study uses the newly developed 10-m global canopy height dataset that merges the Global Ecosystem Dynamics Investigation (GEDI) space-borne LiDAR height data with Sentinel-2 satellite data (Lang et al., 2023). The original 10-m resolution was remapped to 0.125º using the average. Additionally, this study uses a global tree height dataset at 1-km for 2005 using spaceborne lidar (Simard et al., 2011) for complementary analysis in the forests (Appendix B).

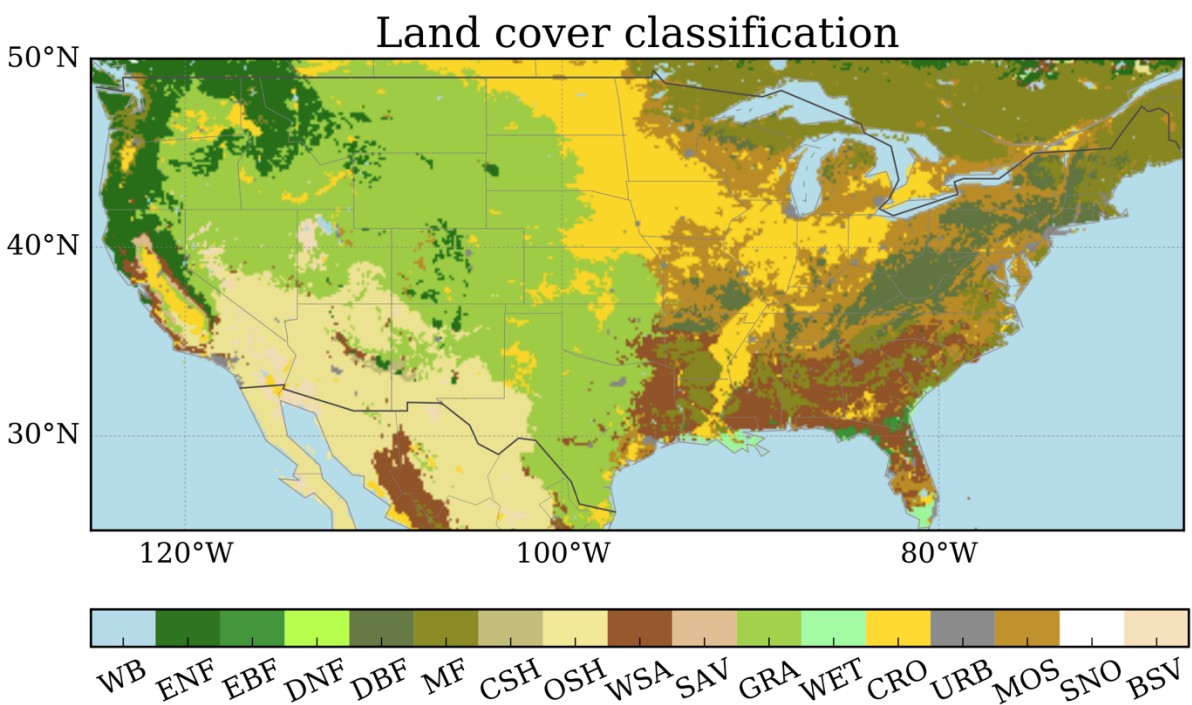

**Figure 1. The land cover classification over the Continental United States used for surface vegetation parameter inference. The classification is based on the satellite retrieval of land cover climatology during 2001-2010 (see Table 1 for a list of land cover full names).**

**3 PET methods**

**3.1 Current PET methods**

PET can be estimated from univariate empirical models such as temperature-based methods (Thornthwaite, 1948) and physically-based models. Empirically based methods can induce large uncertainty in the drought projection (Sheffield et al., 2012; Feng, Trnka, Hayes, & Zhang, 2017) and are therefore not considered in the study. Physically-based methods can account for multiple input variables such as surface net radiation, near-surface temperature, wind speed, or specific humidity. The Penman equation (Penman, 1948) is the most comprehensive physically-based method to estimate PET by combining the radiative and aerodynamic components:

$$PET_{Penman} = \frac{\Delta(R_n - G) + \rho_a C_p D Ga}{\lambda(\Delta + \gamma)} \tag{1}$$

where PET is expressed as water mass fluxes (kg m$^{-2}$ s$^{-1}$), $R_n$ is the surface net radiation (W m$^{-2}$), $G$ is the surface ground heat flux (W m$^{-2}$), $\Delta$ is the slope of the saturation vapor pressure curve at the temperature of interest (Pa K$^{-1}$), $\gamma$ is the psychrometric constant (Pa K$^{-1}$), $\lambda$ is the latent heat of vaporization (J kg$^{-1}$), $\rho_a$ is the air density (kg m$^{-3}$), $C_p$ is the specific heat of air (J kg$^{-1}$ K$^{-1}$), $D$ is the vapor pressure deficit (VPD, Pa), and $Ga$ is the aerodynamic conductance (m s$^{-1}$). The variants of the Penman equation have been widely used to estimate PET in hydrological and land surface modeling (Sellers et al., 1996; Liang et al., 1994; Ek et al., 2003; Peng, Li, & Sheffield, 2018; Peng et al., 2019; Yang et al., 2019).

The open-water Penman (OW) equation is a simplified Penman equation to calculate PET over an open water surface, re-parameterized by Shuttleworth (1993):

$$PET_{OW} = \frac{\Delta}{(\Delta + \gamma)} \frac{(R_n - G)}{\lambda} + \frac{\gamma}{\Delta + \gamma} \frac{6.43(1 + 0.536u_2)D}{\lambda} \tag{2}$$

where $PET_{OW}$ is typically in mm d$^{-1}$ (kg m$^{-2}$ s$^{-1}$ = 86400 mm d$^{-1}$), $(R_n - G)$ is daily available energy (J m$^{-2}$ d$^{-1}$), $u_2$ is the wind speed at 2-m height (m s$^{-1}$), $\lambda$ is J kg$^{-1}$, and $D$ is in kPa. Note that the OW equation provides daily estimates, and therefore some of the variables have different units compared to those in Equation 1.

The Priestley-Taylor (PT) equation is also a simplified form of the Penman equation, which describes evaporation from a well-watered surface based on the equilibrium evaporation under conditions of minimal advection (Priestley & Taylor, 1972):

$$PET_{PT} = 1.26 \frac{\Delta(R_n - G)}{\lambda(\Delta + \gamma)} \tag{3}$$

where $PET_{PT}$ is in mm d$^{-1}$ and $(R_n - G)$ is in J m$^{-2}$ d$^{-1}$.

The Penman-Monteith (PM) equation (Monteith, 1965) is an extended version of the Penman equation to estimate actual ET ($kg\ m^{-2}\ s^{-1}$), which introduces the surface conductance ($Gs$, $m\ s^{-1}$):

$$PET_{PM} = \frac{\Delta(R_n - G) + \rho_a C_p D Ga}{\lambda(\Delta + \gamma\left(1 + \frac{Ga}{Gs}\right))} \tag{4}$$

The reference crop evapotranspiration ($PET_{RC}$) recommended by the UN Food and Agricultural Organization (FAO) is a specific application of the Penman-Monteith equation (Allen, Pereira, Raes, & Smith, 1998). It is designed for calculating the maximum ET of reference crop under well-watered condition. The general formula is given by Allen et al. (2005):

$$PET_{RC} = \frac{0.408\Delta(R_n - G) + \frac{C_n u_2}{T_a + 273}\gamma D}{\Delta + \gamma(1 + C_d u_2)} \tag{5}$$

where $PET_{RC}$ is also in $mm\ d^{-1}$, ($R_n - G$) is daily available energy ($MJ\ m^{-2}\ d^{-1}$), $\Delta$ and $\gamma$ are in $kPa\ °C^{-1}$, $T_a$ is the air temperature at 2-m height (°C), $D$ is in kPa, $C_n$ ($K\ mm\ s^3\ Mg^{-1}\ d^{-1}$) is a constant describing the effect of aerodynamic conductance ($Ga$) that increases with canopy height. The denominator $\Delta + \gamma(1 + C_d u_2)$ is a special form of the denominator of the Penman-Monteith equation $\Delta + \gamma(1 + Rs/Ra)$. $C_d$ ($\frac{Rs}{Ra\ u_2}$, $s\ m^{-1}$) is a constant that increases with the ratio of surface resistance ($Rs = 1/Gs$) to aerodynamic resistance ($Ra = 1/Ga$). There are two sets of $C_n$ and $C_d$, tall crop ($C_n$=1600, $C_d$=0.38) and short crop ($C_n$=900, $C_d$=0.34). The FAO short crop equation is used in the recent version of the SPEI calculation (Beguería, Vicente-Serrano, Reig, & Latorre, 2013).

The above-mentioned equations treat the surface vegetation as a "big leaf" by considering the canopy resistance and soil resistance together as the bulk surface resistance, and therefore require fewer parameters and less computational costs. One challenge of the big-leaf assumption is to infer bulk surface resistance from canopy resistance when the surface is not fully covered by vegetation (Leuning et al., 2008). Additionally, we compare the big leaf models with the Shuttle-Wallace (SW) two source model (Shuttleworth and Wallace, 1985; Sun et al., 2023), incorporating vegetation cover and separating ET into the sum of transpiration and soil evaporation:

$$PET_{SW} = C_c PET_{PMc} + C_s PET_{PMs} \tag{6}$$

where the formulas and parameterizations of $PET_{PMc}$, $PET_{PMs}$, $C_c$, and $C_s$ are given in the Appendix A.

**3.2 Surface characteristics formulas**

Classical PET definitions rely on surface meteorology and do not faithfully represent the vegetation conditions and biophysical constraints and become less accurate over non-uniform land surfaces (Moran et al., 1996). This section introduces the major options of formulas for aerodynamic conductance and surface conductance.

### 3.2.1 Aerodynamic conductance

Aerodynamic conductance $Ga$ in the OW and PT methods (Equations 2 and 5) are implicitly derived from a smooth surface with low roughness length, which can underestimate the $Ga$ and PET values in the forests (Peng et al., 2019). Open water aerodynamic conductance $Ga_{OW}$ can be obtained by inverting the open water Penman equation (Equation 2) to match the Penman equation (Equation 1), given by Peng et al. (2019):

$$Ga_{OW} = \frac{6.43(1 + 0.536u_2) \cdot P_s}{86.4\epsilon\lambda\rho_a} \tag{6}$$

where $u_2$ is converted from wind speed at 10-m to 2-m height following the wind profile relationship in Allen, Pereira, Raes, & Smith (1998). $P_s$ is near-surface atmospheric pressure (Pa), $\epsilon$ is the ratio of molecular weight of water to dry air (= 0.622).

Short and tall reference crop aerodynamic conductance $Ga_{RC-short}$ and $Ga_{RC-tall}$ are given by

$$Ga_{RC-short} = \frac{u_2}{208} \tag{7}$$

$$Ga_{RC-tall} = \frac{u_2}{110} \tag{8}$$

where $u_2$ is converted from wind speed at 10-m to 2-m height (m s$^{-1}$).

Instead of the low $Ga$ in OW and the fixed $Ga$ in RC, it is better to generate more realistic surface roughness varying by land cover type, hereafter called $Ga_{LC}$ (Brutsaert & Stricker, 1979; Allen, Pereira, Raes, & Smith, 1998; Shuttleworth, 1993):

$$Ga_{LC} = \frac{k^2 u_z}{\ln\left(\frac{z_m - d_0}{z_{0m}}\right)\ln\left(\frac{z_h - d_0}{z_{0h}}\right)} \tag{9}$$

where $z_m$ is the measurement height (m) for wind speed, $z_h$ is the measurement height (m) for temperature and humidity, $u_z$ is the wind speed at measurement height (m s$^{-1}$), $k$ is the von Karman constant, $d_0$ is the zero-plane displacement height (m), $z_{0m}$ and $z_{0h}$ are the roughness lengths for momentum and heat (m). $d_0$ and $z_{0m}$ can be estimated from canopy height (h) following $d_0 = 2h/3$ and $z_{0m} = h/8$ (Brutsaert, 1982). $h$ is

based on the typical value for each land cover type. When estimating $z_{0h}$, instead of assuming $z_{0h} = 0.1z_{0m}$ as in Allen, Pereira, Raes, & Smith (1998), it is common to introduce a concept of excess resistance (Verma, 1989) and characterize the relationship between $z_{0h}$ and $z_{0m}$:

$$z_{0h} = \frac{z_{0m}}{\exp(kB^{-1})} \qquad (10)$$

The $\ln(z_{0m}/z_{0h})$ term, also known as $kB^{-1}$, depends on the roughness Reynold's number $Re*$ or frictional velocity ($u*$), LAI (Yang & Friedl, 2003), and land cover type (Rigden, Li, & Salvucci, 2018).

On top of the above land cover based roughness (Equation 9), it is possible to further incorporate realistic canopy height ($h$) to account for its effect on wind speed, hereafter called $Ga_{CH}$:

$$Ga_{CH} = \frac{k^2 u_r}{\ln\left(\frac{z_r - d_0}{z_{0m}}\right) \ln\left(\frac{z_r - d_0}{z_{0h}}\right)} \qquad (11)$$

Similar to Equation 9, $d_0$ and $z_{0m}$ are estimated from canopy height (h) following $d_0 = 2h/3$ and $z_{0m} = h/8$ (Brutsaert, 1982), and $z_{0h}$ is estimated using Equation 10; the only difference is that this CH approach use actual canopy height data instead of look up table. Different from Equation 9, this approach assumes a reference level, where $z_r$ is the reference height (2m above canopy height) for wind speed, temperature, and humidity (Zhou et al., 2006). The reference height wind speed $u_r$ (m s$^{-1}$) is converted from the measured wind speed $u_z$ following the wind profile relationship. The internal boundary layer ($z_b$) on top of the measurement height ($z$ = 10m) and canopy reference height are matched (Zhou et al., 2006; Brutsaert, 1982; Federer et al., 1996):

$$u_r = u_z \frac{\ln\left(\frac{z_b}{z_{0g}}\right) \ln\left(\frac{z_r - d_0}{z_{0m}}\right)}{\ln\left(\frac{z_b}{z_{0m}}\right) \ln\left(\frac{z}{z_{0g}}\right)} \qquad (12)$$

where ground roughness length $z_{0g}$ is 0.005 m, $z_r$ is the reference height at ($h+2$) m, $z$ is the measurement height at 10 m. The internal boundary layer height $z_b$ is estimated at about 4.4 m:

$$z_b = 0.334F^{0.875}z_{0g}^{0.125} \qquad (13)$$

where $F$ is the fetch at 5000 m, the effective distance over which the wind blows do not change in direction.

For the SW method, the two aerodynamic resistances are given by Eq. A11-17 (Appendix A).

### 3.2.2 Surface conductance

In previous PET methods, surface conductance is either not considered or assumed to be constant across vegetation types and over time. LAI plays a dominant role in determining the canopy-atmosphere coupling and ET partitioning (Peng et al, 2019; Wei et al., 2017; Forzieri et al., 2020). The OW and PT approach does not consider the role of LAI. The FAO approach uses a constant LAI throughout the growing season. Here we adopt a widely used method in estimating actual ET and assume a well-watered condition. The maximum surface conductance $Gs_{max}$ can be obtained by scaling the maximum stomatal conductance ($Gst_{max}$) with LAI (Yan et al., 2012):

$$Gs_{max} = Gst_{max} \cdot LAI \tag{14}$$

An alternative formula for $Gs_{max}$ is from Zhou et al. (2006):

$$Gs_{max} = \frac{LAI_e}{Rst_{min}} \tag{15}$$

where $LAI_e$ is the effective LAI, which is equal to LAI/2 when LAI is greater than 4. We introduce two options to incorporate an average LAI or the seasonal cycle of LAI into the surface conductance.

### 3.3 Parameterizations of surface characteristics

For Eq. 9, given that NLDAS-2 provides wind speed at a 10 m level, we used a measurement height = 10 m for both wind speed and temperature because the variation in the vertical temperature profile (2-10 m) is negligible compared to that of wind speed. For $z_{0m}$, we apply the typical values based on median canopy height for different land cover types, and estimated $d_0$ from $z_{0m}$ ($d_0 \approx 16z_{0m}/3$). We use a simple look-up table approach to provide parameters based on land cover type (Fig. 1), summarized in Table 1.

For $kB^{-1}$, we adopt estimates from a collection of literature as below. The forests generally have lower $kB^{-1}$ values ($kB^{-1} = 1$ for needleleaf or mixed forest, $kB^{-1} = 0.5$ for broadleaf) than shrublands ($kB^{-1} = 3.75$) and croplands ($kB^{-1} = 1.75$), based on the values of Rigden et al. (2018) for the medium emissivity case ($\epsilon = 0.96$). For grasslands, $kB^{-1} = 2.25$ is computed as the average of short grass ($kB^{-1} = 2.0$) and medium-length grass ($kB^{-1} = 2.5$), based on Brutsaert (1982). For barren or bare soil, we estimate $kB^{-1} = 3$ by taking the average of all observed $kB^{-1}$ in Yang et al. (2008). Nadeau et al. (2009) suggested $kB^{-1} = 6$ for the urban area. For water body, wetlands, and snow, we adopt the widely-used $kB^{-1} = 2$, as Zilitinkevich et al. (2001) showed that $kB^{-1}$ over the water surface is within the 0-4 range. There are large variations in the observed $kB^{-1}$ for savannas. Troufleau et al. (1997) reported $kB^{-1} = 7.9$ for fallow savanna; Kustas et al.

(1989) provided a range of 1 to 11; Stewart et al. (1994) found an average value of $kB^{-1} = 5.8$, similar to the study by Lhomme et al. (1997) that reported $kB^{-1} = 5.9$ for Sahelian vegetation; Verhoef et al. (1997) suggested a high value of $kB^{-1} = 12.4$. We choose $kB^{-1} = 7$ as most of these observed values fall into the range of 6-8. $z_{0h}$ is then estimated based on land cover specific $z_{0m}$ and $kB^{-1}$ (Eq. 10).

**Table 1. $Ga$ and $Gs$ parameters by IGBP land cover\*.**

| ID | Code | Name | $z_{0m}$ (m) | $d_0$ (m) | $kB^{-1}$ | $Gst_{max}$ [j] (mm s$^{-1}$) | $Rst_{min}$ [k] (s m$^{-1}$) |
|---|---|---|---|---|---|---|---|
| 0 | WB | Water body | 0.0004 [a] | 0.002 | 2.0 [e] | NA | NA |
| 1 | ENF | Evergreen needleleaf | 1.1 [b] | 5.9 | 1.0 [f] | 9.3 | 150 |
| 2 | EBF | Evergreen broadleaf | 1.1 [b] | 5.9 | 0.5 [f] | 9.3 | 150 |
| 3 | DNF | Deciduous needleleaf | 0.9 [b] | 4.8 | 1.0 [f] | 9.3 | 150 |
| 4 | DBF | Deciduous broadleaf | 0.9 [b] | 4.8 | 0.5 [f] | 9.3 | 150 |
| 5 | MF | Mixed forest | 0.9 [b] | 4.8 | 1.0 [f] | 9.3 | 150 |
| 6 | CSH | Closed shrublands | 0.2 [a] | 1.1 | 3.75 [f] | 9.3 | 150 |
| 7 | OSH | Open shrublands | 0.2 [a] | 1.1 | 3.75 [f] | 9.3 | 100 |
| 8 | WSA | Woody savannas | 0.4 [a] | 2.1 | 7.0 [g] | 9.3 | 180 |
| 9 | SAV | Savannas | 0.4 [a] | 2.1 | 7.0 [g] | 9.3 | 120 |
| 10 | GRA | Grasslands | 0.05 [a] | 0.27 | 2.25 [a] | 12 | 115 |
| 11 | WET | Permanent wetlands | 0.04 [c] | 0.21 | 2.0 [e] | 12 | 65 |
| 12 | CRO | Croplands | 0.12 [d] | 0.64 | 1.75 [f] | 12.2 | 90 |
| 13 | URB | Urban and built up | 1.1 [b] | 5.9 | 6.0 [h] | NA | NA |
| 14 | MOS | Cropland/vegetation | 0.12 [d] | 0.64 | 1.75 [f] | 12.2 | 120 |
| 15 | SNO | Snow/ice | 0.00001 [a] | 5.3E-05 | 2.0 [e] | NA | NA |
| 16 | BSV | Barren | 0.01 [d] | 0.053 | 3.0 [i] | NA | NA |

\*The above estimates are collected from [a]Brutsaert (1982), [b]Campbell and Norman (1998), [c]Acreman et al. (2003), [d]Monteith and Unsworth (2013), [e]Zilitinkevich et al. (2001), [f]Rigden et al. (2018), [g]Kustas et al. (1989), Stewart et al. (1994), Troufleau et al. (1997), Lhomme et al. (1997), and Verhoef et al. (1997), [h]Nadeau et al. (2009), [i]Yang et al. (2008), [j]Kelliher et al. (1995), [k]Zhou et al. (2006).

Canopy height ($h$) is a key parameter in determining aerodynamic conductance and is eventually used to obtain $d_0$ and $z_{0m}$ for Eq.9. The OW and FAO methods generally assume it to be constant across vegetation types and temporal scales. To address this limitation, we evaluate two methods for canopy height parameterization.

The first method uses literature values and is adopted in the Land Cover approach (LC, Eq.9). For most of the land cover types (ID 6-16), we applied the values from the look up table except for the forests, where we

determined canopy height by calculating the median height within each land cover from the tree height lidar data (Simard et al., 2011).

The second more comprehensive method is adopted in the Canopy Height approach (CH, Eq. 11) and the SW two source model (Appendix A, Eq. A9-10). It takes into account three factors: land cover type, measured canopy height, and dynamic LAI. We overlaid the land cover map (Fig. 1) and the canopy/tree height data (Lang et al., 2023; Simard et al., 2011) to obtain the distribution in each land cover type (Appendix B). Based on the distribution of the two datasets, land cover definition, and literature ranges, we estimated the minimum canopy height ($h_{min}$) and maximum canopy height ($h_{max}$) by land cover type (Table 2). As for quality control, we set the outlier (smaller than $h_{min}$ or greater than $h_{max}$) to a typical value of canopy height given land cover type ($h_{typ}$, obtained through the mode of the distribution). The actual canopy height is then determined by assuming a linear relationship with dynamic LAI following Zhou et al. (2006).

$$h = h_{min} + \frac{(h_{max} - h_{min})LAI}{LAI_{max}} \tag{13}$$

where $LAI_{max}$ represents the annual maximum value at the grid cell level, obtained from the satellite data. Note that h is set to zero if $LAI_{max}$ is zero.

**Table 2. Canopy height parameters by IGBP land cover*.**

| ID | Code | Name | $h_{min}$ (m) | $h_{max}$ (m) | $h_{typ}$ (m) |
|----|------|------|------|------|------|
| 0 | WB | Water body | 0.001 | 0.02 | 0.01 |
| 1 | ENF | Evergreen needleleaf | 2 [a] | 48 [b] | 13 [b] |
| 2 | EBF | Evergreen broadleaf | 2 [a] | 45 [b] | 17 [b] |
| 3 | DNF | Deciduous needleleaf | 7 [b] | 23 [b] | 17 [b] |
| 4 | DBF | Deciduous broadleaf | 6 [b] | 37 [c] | 9.5 [c] |
| 5 | MF | Mixed forest | 2 [a] | 32 [b] | 25 [b] |
| 6 | CSH | Closed shrublands | 1 [b] | 39 [b] | 14.9 [b] |
| 7 | OSH | Open shrublands | 2 [b] | 17 [b] | 6 [b] |
| 8 | WSA | Woody savannas | 1 [b] | 23 [b] | 1 [b] |
| 9 | SAV | Savannas | 1 [b] | 26 [b] | 17.7 [b] |
| 10 | GRA | Grasslands | 0.1 | 3 | 1.5 |
| 11 | WET | Permanent wetlands | 0.1 | 5 | 0.5 |
| 12 | CRO | Croplands | 0.1 | 5 | 1 |
| 13 | URB | Urban and built up | 2 | 50 | 13 |
| 14 | MOS | Cropland/vegetation | 0.1 | 21 | 12 |
| 15 | SNO | Snow/ice | 0.001 | 0.02 | 0.01 |
| 16 | BSV | Barren | 0.01 | 0.1 | 0.05 |

*The above estimates are collected from [a]IGBP classification, [b]Lang et al. (2023), [c]Simard et al. (2023), otherwise based on authors' best estimates.

To calculate surface conductance in Eq. 11-12, we provide two set of parameterizations based on land cover type. The first set is derived from the findings of Kelliher, Leuning, Raupach, & Schulze (1995). For $Gst_{max}$, the measured values are ranging from 9 mm/s for natural vegetation to 12 mm/s for crops, as detailed in Table 1. They also found that $Gs_{max}$ estimates are at most three times of the $Gst_{max}$ estimates, therefore we set a maximum limit for $LAI$ = 4. The second set uses the minimum stomatal resistance $Rst_{min}$, following Zhou et al. (2006), also listed in Table 1.

Current PET methods generally apply a uniform grass albedo value of 0.23 regardless of the underlying land cover type (Allen, Pereira, Raes, & Smith, 1998). To improve upon this assumption, we also introduce an option of introducing seasonal albedo cycle from satellite observations to both align albedo with specific land cover type and reflect temporal variations accurately.

## 4 Evaluation of the PET methods and parameterizations

### 4.1 Drought quantification: SPEI vs. soil moisture

Given the substantial divergence in the PET magnitudes among different models (Peng et al., 2019), a direct comparison of the absolute values among methods is not meaningful. However, the performance in representing drought between PET methods should be comparable. We hypothesize that incorporating the parameters or model structures in Section 3 into the existing methods will increase the accuracy of drought quantification.

We integrate the PET methods into the SPEI drought index across 1-, 3-, 6-, and 12-month time scales over the CONUS for the period of 1981-2017. The SPEI is based on the climatological water balance (water supply – atmospheric evaporative demand) cumulated over multiple time scales (e.g., 1, 3, 6, 12 months) following a similar procedure as in the SPI computation (Vicente-Serrano, Beguería, & López-Moreno, 2010). The accumulated water balances are fit using the log-logistic distribution and the probability distribution function is normalized to a standardized variable with mean = 0 and standard deviation = 1, termed as 1-, 3-, 6-, 12-month SPEI, respectively. We calculate the monthly SPEI with the SPEI R package (https://cran.r-project.org/web/packages/SPEI/) using daily meteorological data. We choose the SPEI driven by zero PET as a control scenario to showcase the net effect of introducing existing PET methods into traditional SPI drought index. We choose the SPEI driven by the Open Water (OW) method as the reference method, because the OW approach is the simplest scenario with minimal surface characteristics.

Soil moisture is a direct measure of drought severity. Therefore, we use the correlation between SPEI and soil moisture observations to quantify the skill of PET methods. We aggregated the daily ESA CCI surface soil moisture (SMsurf, $m^3 \ m^{-3}$) to monthly averages between 1981-2017 over the CONUS. To match the SPEI on multiple time scales, we calculated the moving average of SMsurf for 1, 3, 6, 12-month periods, respectively. Our analysis focuses on the growing season (April-September), because PET is close to zero during the cold season (not shown). Given the monthly SPEI and SMsurf series during the growing season, Pearson correlation coefficient ($R$) is calculated for each pair of the SPEI and SMsurf monthly series in each grid cell of the CONUS on the time scale of 1, 3, 6, and 12 months. Then we calculate the change of correlation for each method from the control scenario or the reference. This change can identify whether a PET method causes an improvement in drought quantification relative to the reference approach.

**4.2 Initial examination of surface characteristics**

We conducted a pilot analysis to identify the relative importance of different surface characteristics. To test the hypotheses, we use the PM algorithm for big leaf methods, so that we can easily control a specific set of parameters that represents a process option. Each of the above processes are regarded as different options: (i) using active surface roughness or open water surface, (ii) seasonally varying or fixed surface conductance, and (iii) seasonally varying or fixed surface albedo. Table 3 provides the PET methods and the parameters in the preliminary analysis. We selected four existing PET methods and seven testing methods. The first set of methods (a, e, i, j) are the existing physically-based PET approaches: the open-water Penman equation (OW), the Priestley-Taylor equation (PT), the FAO reference crop evapotranspiration for short and tall crop.

First, in methods (b-d), the aerodynamic conductance module is not active as we set $Ga$ to the open water $Ga_{OW}$, indicating a smooth surface with low roughness (Eq. 6). In methods (f-h), we activate the aerodynamic conductance using land cover based surface roughness as $Ga_{LC}$ (Eq. 9). In methods (k-l), we activate the aerodynamic conductance using the formula of reference short crop (Eq.7). Second, in methods (b), (g), (k), the surface conductance parameter is unconstrained as we set $Gst_{max}$ to infinity ($Gs_{OW}$). In methods (c), (d), (g), (h), (l), we activate the surface conductance using seasonal LAI dynamics and $Gst_{max}$ from Kelliher et al. (1995). Lastly, in methods (c), (g), (k), (l), the albedo parameter is not active as we set $\alpha$ to a constant (RC: 0.23 for grass, OW: 0.08 for water). In methods (b), (d), (f), (h), we activate the albedo parameter using seasonal albedo dynamics ($\alpha_{CLM}$).

**Table 3. Summary of the PET methods for initial assessment with their ID, name and abbreviation code, and details about surface characteristics.**

| ID | Method (Code) | $Ga$ | | $Gs_{max}$ | | | Albedo ($\alpha$) | |
|---|---|---|---|---|---|---|---|---|
| | | Open Water | Rough surface | Infinite | Constant | Seasonal | Constant | Seasonal |
| a | Open Water (OW) | X | | X | | | X | |
| e | Priestley-Taylor (PT) | | | | | | X | |
| i | FAO Short reference crop (RC-short) | | X | | X | | X | |
| j | FAO Tall reference crop (RC-tall) | | X | | X | | X | |
| b | Ga OW\|Gs OW\|α CLM | X | | X | | | | X |
| c | Ga OW\|Gs LAI\|α OW | X | | | | X | X | |
| d | Ga OW\|Gs LAI\|α CLM | X | | | | X | | X |
| f | Ga LC\|Gs OW\|α CLM | | X | X | | | | X |
| g | Ga LC\|Gs LAI\|α OW | | X | | | X | X | |
| h | Ga LC\|Gs LAI\|α CLM | | X | | | X | | X |
| k | Ga RC\|Gs OW\|α RC | | X | X | | | X | |
| l | Ga RC\|Gs LAI\|α RC | | X | | | X | X | |

*Note that many methods in these experiments are unrealistic due to the inconsistencies of the surface conditions. Our attention is to include as many combinations as possible for a preliminary analysis.

355 In Section 5.1, we compare the CONUS averaged $R$ values between the pairs of PET methods that share the same surface characteristics except for one of the features (see Fig. 3). The first feature surface roughness is determined by the way $Ga$ is estimated. We compare the parameter set between rough and the open water surface by calculating the differences (Rough - Open Water) for the following pairs of experiments including (f) Ga LC|Gs OW|α CLM – (b) Ga OW|Gs OW|α CLM, (g) Ga LC|Gs LAI|α OW – (c) Ga OW|Gs LAI|α

360 OW, and (h) Ga LC|Gs LAI|α CLM – (d) Ga OW|Gs LAI|α CLM. In terms of the surface conductance, we calculate the differences between seasonal and infinite $Gs_{max}$ (Seasonal - Infinite) for the following pairs of experiments: (c) Ga OW|Gs LAI|α OW – (a) OW, (d) Ga OW|Gs LAI|α CLM – (b) Ga OW|Gs OW|α CLM, and (h) Ga LC|Gs LAI|α CLM – (f) Ga LC|Gs OW|α CLM. We also compare the differences between all consistent and inconsistent surfaces and between seasonal and constant albedo.

365 **4.3 Comparison of PET parameterizations**

Based on the results of section 5.1, we further examine different parameterizations for $Ga$ and $Gs$ in order to identify optimal PET algorithms (Table 4, results in Section 5.2). We establish a control scenario where PET is not considered at all in the SPEI, equivalent to the traditional SPI. The PET methods under consideration have two categories, the big leaf model and the two source model. The big leaf model include three traditional

370 methods (Open Water [OW], Reference Crop for short [RC-short], and tall [RC-tall] crops), land cover dependent (LC), and canopy height dependent (CH) methods. The LC method uses the same aerodynamic conductance method (Eq. 9) but differ in their surface conductance parameterizations: LC-K, which adopts

$Gst_{max}$ from Kelliher, Leuning, Raupach, & Schulze (1995), and LC-Z, which uses $Rst_{min}$ from Zhou (2006). The CH method also has the two parameterizations: CH-K and CH-Z. We then calculated $\Delta R$ between each PET method and the control scenario (set PET to zero).

**Table 4. Summary of PET methods with their formula and parameterization**

| type | Model code (equation) | Ga formula | Ga parameter | Gs formula | Gs parameter |
|---|---|---|---|---|---|
| | OW (Eq. 2) | Eq. 6 | Peng et al. 2019 | - | - |
| | PT (Eq. 3) | - | - | - | - |
| | RC-short (Eq. 5) | Eq. 7 | Allen et al. 2005 | Eq. 5 | Allen et al. 2005 |
| Big leaf | RC-tall (Eq. 5) | Eq. 8 | Allen et al. 2005 | Eq. 5 | Allen et al. 2005 |
| | LC-K (Eq. 4) | Eq. 9-10 | $z_{0m}$ (m), $d_0$ (m), $kB^{-1}$ (Table 1); Tree height (Simard et al. 2011) | Eq. 14 | Kelliher et al. 1995 |
| | LC-Z (Eq. 4) | | | Eq. 15 | Zhou et al. 2006 |
| | CH-K (Eq. 4) | Eq. 11-13 | Canopy height data (Lang et al. 2023) | Eq. 14 | Kelliher et al. 1995 |
| | CH-Z (Eq. 4) | | | Eq. 15 | Zhou et al. 2006 |
| Two source | SW (Eq. A1-5) | Eq. A6 | Zhou et al. 2006; Canopy height data (Lang et al. 2023) | Eq. A7-8 | Zhou et al. 2006 |

## 5 Results

### 5.1 Initial assessment of surface characteristics

We conducted a preliminary analysis to identify the relative importance of different surface characteristics. We examine eight algorithms to isolate the effects of surface characteristics on PET (Table 2). Fig. 2 displays the spatial patterns of growing season averages of these methods. For the classical Penman/Penman-Monteith methods (Fig. 2a, i, j), the highest mean growing season AED values are found in southern California, Arizona, and Texas, while the PT method (Fig. 2e) predicts the largest AED values in Texas and Florida. The spatial patterns of PET based on the rough surface (Fig. 2f-h, Rough $Ga$) are very different from those methods that assume a universal reference height (Fig. 2i-j, reference crop) or open water surface (Fig. 2a, b-d). Specifically, the regions which exhibit large PET estimates ($> 250$ mm/mon, Fig. 2h-k) are forests, such as ENF in the Pacific Northwest, DBF in the Northeast, and MF in the southeastern U.S.. Interestingly, although the methods using constant albedo ($\alpha$=0.08) have generally larger AED values than those using seasonal albedo, the differences in the spatial pattern between the two are almost negligible (Fig. 2c vs. d, g vs. h). The combination of the rough aerodynamic and unconstrained surface conductance, represented by (f), produces extremely high monthly PET values with means at 330 to 340 mm/mon. The remaining methods also predict a wide range of mean monthly totals.

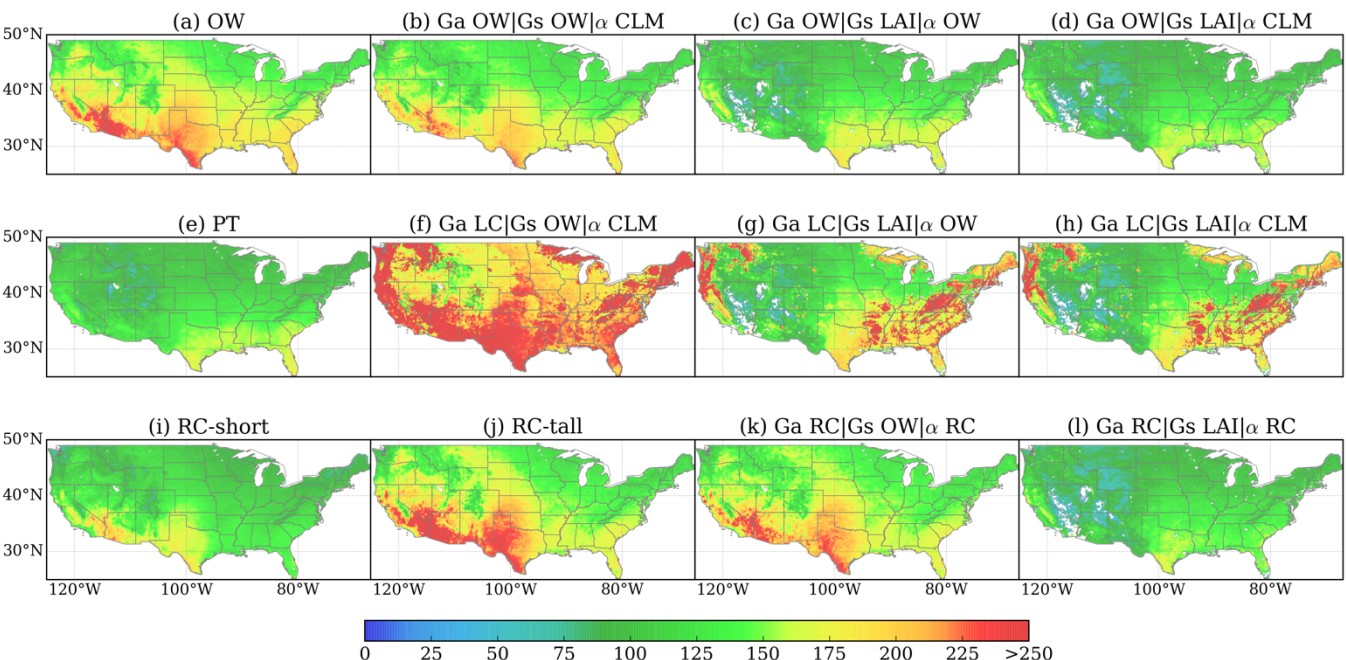

**Figure 2. Growing season averages of AED derived from four PET methods and eight testing algorithms over the CONUS. Details and ID for each method are listed in Table 2.**

Assessing the change between pairs of the above methods can identify whether adding/removing a surface feature eventually causes an improvement in drought quantification (Fig. 3). For 1-month time scale, surface roughness stands out to be the most important feature for enhancing the skill of drought index ($\Delta R$ = 0.01-0.025). For 6-month time scale, interestingly, activating realistic surface roughness does not necessarily

increase the correlation with SMsurf, while activating dynamic surface conductance improves the correlation, meaning that adding the plant phenology driven by LAI can improve the seasonal variations of drought index over longer time scale. We compare $\Delta R$ of five pairs with an inconsistent surface (e.g., a combination of open water $Ga$ and seasonal $Gc$) subtracting from a consistent surface and find that methods with consistent surface features have persistently higher correlations with SMsurf ($\Delta R$ = 0 - 0.02). Given the steadily better

performance, we only focus on the consistent surface approaches in subsequent sections. Surprisingly, seasonal and constant albedo showed no significant difference on the correlations, possibly because of the little variation of albedo during the growing season. The differences in the spatial pattern between constant and seasonal albedo are almost negligible (Fig. 2). In subsequent sections, we default to using the seasonal albedo in our PET methods to fully represent the surface characteristics.

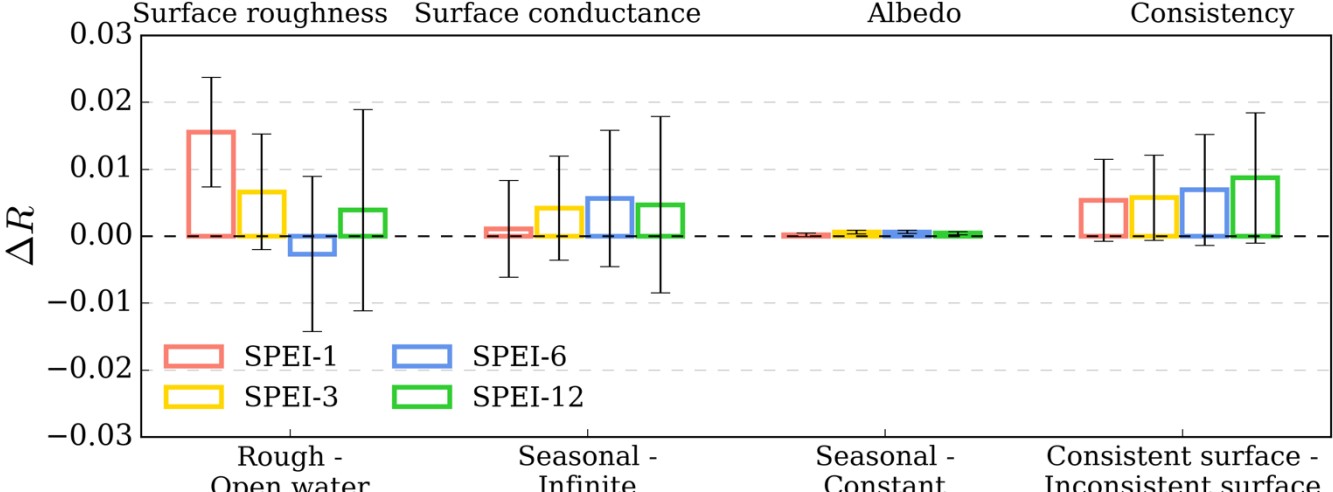

**Figure 3. Differences in spatially averaged correlation ($\Delta R$) of pairs of PET methods that share the same surface characteristics except for one of the surface features: surface roughness, surface conductance, albedo, and overall consistency among the above features.**

## 5.2 Performance of PET parameterizations

Fig. 4 shows ΔR between each PET method in Table 4 and the control scenario (set PET to zero) for all grids, forested grids, and nonforested grids using 1-month SPEI. Incorporating the benchmark OW method into the SPEI increases $R$ by 0.042, shown by the top horizontal bars. Among the conventional PET methods, the tall reference crop (RC-tall) method stands out. Over the CONUS, it improved ΔR relative to control scenario by 29% more than the OW method (0.054 versus 0.042). The short reference crop (RC-short) method has an identical averaged $R$ with the OW method. Although the RC-tall algorithm (Allen et al., 2005) is less known than the widely used RC-short algorithm (Allen, Pereira, Raes, & Smith, 1998), our results suggest that the SPEI driven by RC-tall correlates better with the SMsurf dynamics.

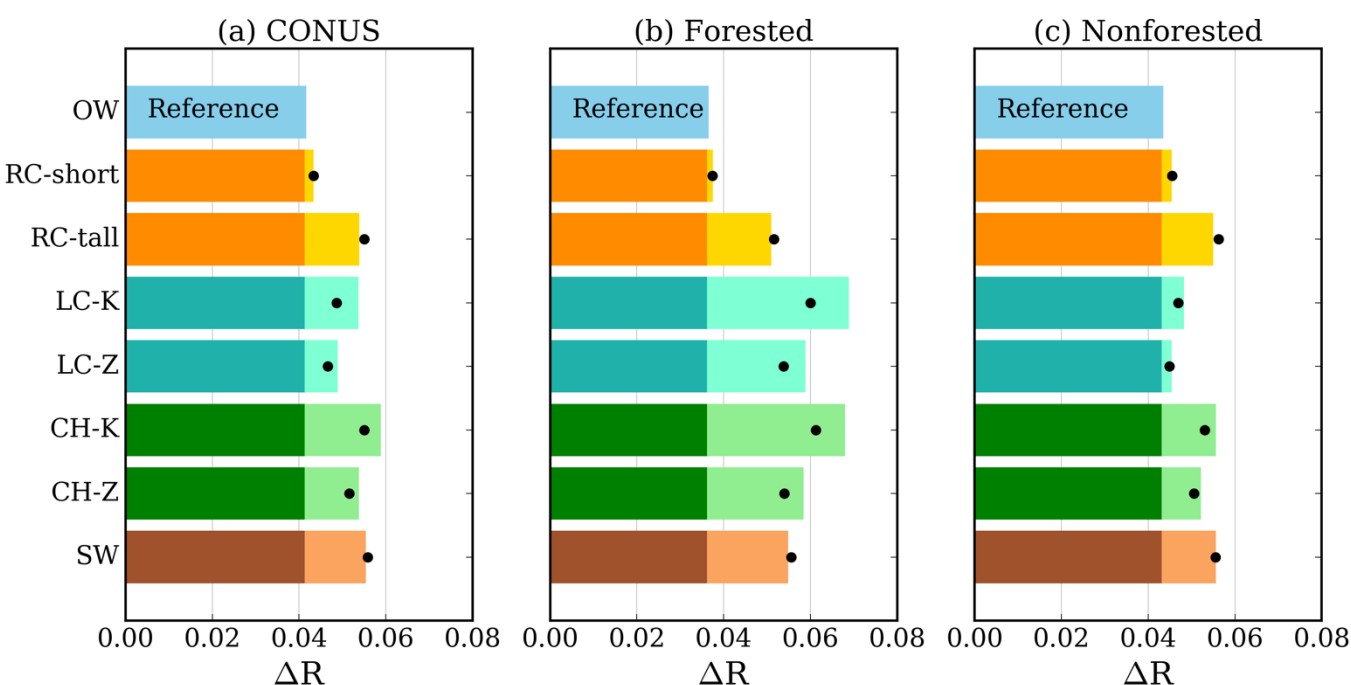

**Figure 4. Differences in correlations (ΔR) for selected PET methods versus the control scenario (PET = 0). Correlations were computed between the 1-month SPEI and SMsurf series across: (a) CONUS, (b) forested grids, and (c) nonforested grids. The bars represent the mean ΔR and the black dots represent the median ΔR. The top blue bars show ΔR in the OW approach versus PET = 0 as a reference. For each bar, the darker shade indicates the reference ΔR and the lighter shade represents any improvement (or decline) relative to the reference.**

One encouraging outcome is the performance improvement seen in the two big leaf LC and CH algorithms incorporating realistic surface conductance. Activating both surface roughness and seasonal $Gs$ produces high correlations of SPEI with SMsurf. These algorithms improve the OW method ($\Delta R = 0.042$) by 29-41% ($\Delta R$ is 0.053-0.059). Methods where $Ga$ is determined by canopy height (the green bars in Fig. 3a) especially improve the correlations with SMsurf. Methods where $Gs$ is determined by Eq. 14 and parameterized using Kelliher et al. produce higher correlations in both LC and CH algorithms. It confirms our hypothesis that incorporating realistic vegetation information in atmospheric evaporative demand can enhance drought characterization. Finally, the two source Shuttleworth-Wallace (SW) method outperforms the OW method as expected. However, the SW method produces a smaller $R$ than the CH-K method. This suggests that the simple big leaf model in combination with the land cover details can achieve the same efficacy of the more complicated two source model.

Over the CONUS, RC-tall, LC-K, CH-K, and SW are the top methods with similar average $R$. However, when we evaluate the performance in the forested areas (Fig. 3b), LC-K and CH-K exhibit the most significant enhancement in $\Delta R$ to control scenario, with increases of 86-89% over OW's improvement (0.068 relative to 0.036). RC-tall and SW improve $\Delta R$ to control scenario by 39% (0.05) and 50% (0.054), respectively. In nonforested areas (Fig. 3c), RC-tall has the best performance, followed by SW and CH-K. The SW method, designed for sparse vegetation, naturally demonstrates strong performance in these regions. Similarly, the CH-K method uses the dataset by Lang et al., which includes better quality canopy height measurements in short vegetated areas. Conversely, LC-K only exhibits a moderate improvement in $\Delta R$. This suggests that the performance of the land cover based approach in the sparse vegetation is strongly influenced by the uncertainty of the roughness parameters. On the other hand, it is surprising to see that the simple parametrized RC-tall can outperform SW. This suggests that, particularly in the sparsely vegetated areas, RC-tall can serve as a strong yet simple approach for PET estimates and drought characterization.

**5.3 Spatial patterns analysis**

In the subsequent sections, we compare the LC-K (referred to as LC) and CH-K (referred to as CH) methods, RC-tall, and SW with the three widely used methods: OW, PT, RC-short. The time series of these PET methods as well as the SMsurf time series are shown in the Fig. 5. The spatial patterns of the mean PET monthly values are shown in Fig. C1 (Appendix C). The OW approach serves as the reference. The highest $R$ is observed for long-term drought (12-month, average $R = 0.73$) and the lowest is found in medium-term

drought (3- and 6-month, average $R = 0.48$). This suggests that the meteorology-driven SPEI can generally reproduce soil moisture dynamics, especially on an annual time scale.

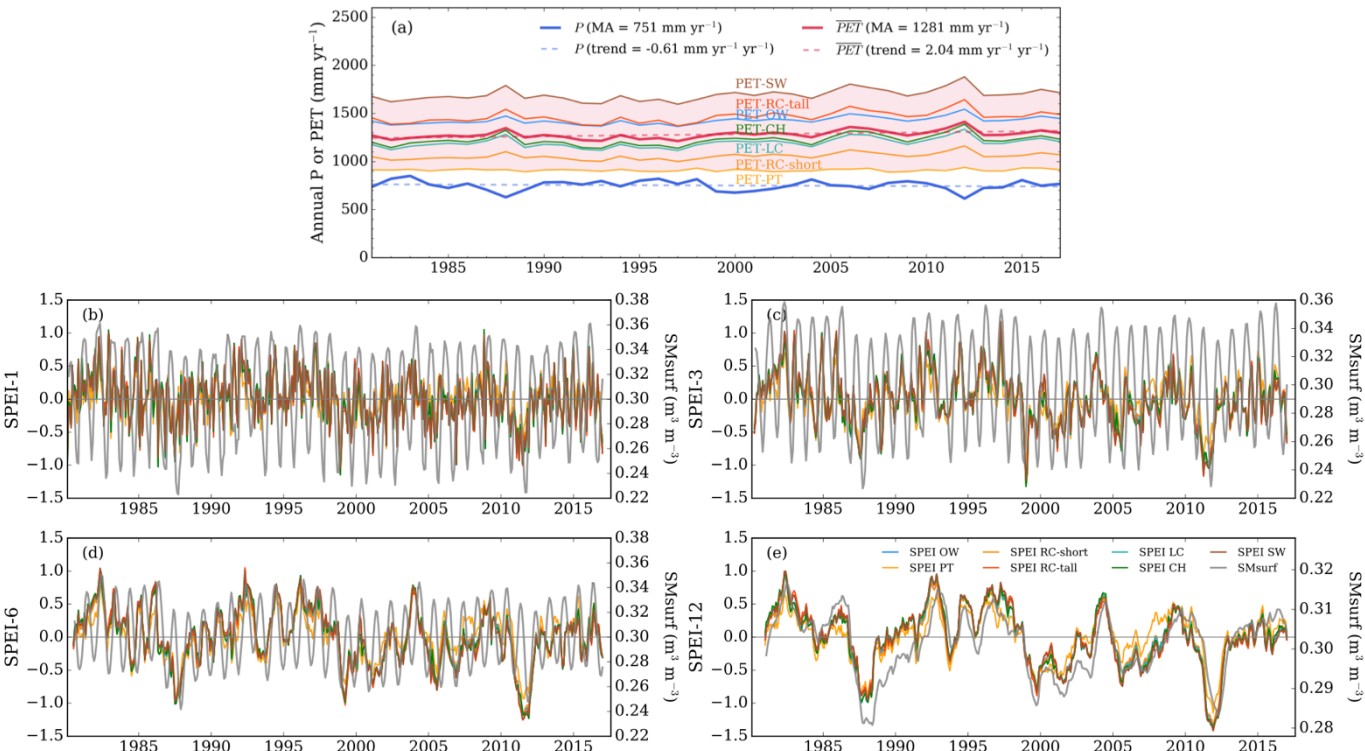

**Figure 5. Temporal evolution of PET methods, SPEIs, and SMsurf. a) The annual precipitation and PET (mm yr⁻¹) from PET methods between 1981-2017. b)-e) SPEI series**

**driven by the PET methods, aligning with the SMsurf time series for four time scales: 1, 3, 6, 12-month.**

Fig. 6 displays the spatial distribution of correlations between SPEI driven by OW and SMsurf, along with the differences in correlations of PT, RC-short, RC-tall, LC, CH, and SW compared to OW. PT consistently

exhibits lower correlations than OW over most regions, with an average decrease of 0.04, and has especially weak correlations in the southwest U.S. (lower by 0.15). Interestingly, the widely used RC-short method for SPEI presents little improvement over OW with minimal increases in correlation, while RC-tall method has overall better performance across CONUS and time scales. Both CH and LC show substantial improvements in some areas, with $\Delta R$ exceeding 0.16, notably in the eastern and pacific western U.S.. The enhancements

of LC and CH are prominent but can be diluted when averaged across CONUS, with $\Delta R$ relative to the control scenario 0.012 higher than OW (Fig. 4a). This is especially true when considering their less favorable performance in the wouthwest and midwest U.S.. SW also exhibits notable improvements in the eastern and

pacific western U.S., with a magnitude of improvement falling between CH and RC-short. It is encouraging to see that LC and CH outperform SW in many eastern US grid cells ($\Delta R = 0.15$ versus $\Delta R = 0.05$), given their much simpler parameterization. Though it is worth noting that all LC, CH, and SW experience performance declines in the Southwest, with LC and CH slightly worse than SW. On the other hand, RC-tall robustly displays improvements in this particular area.

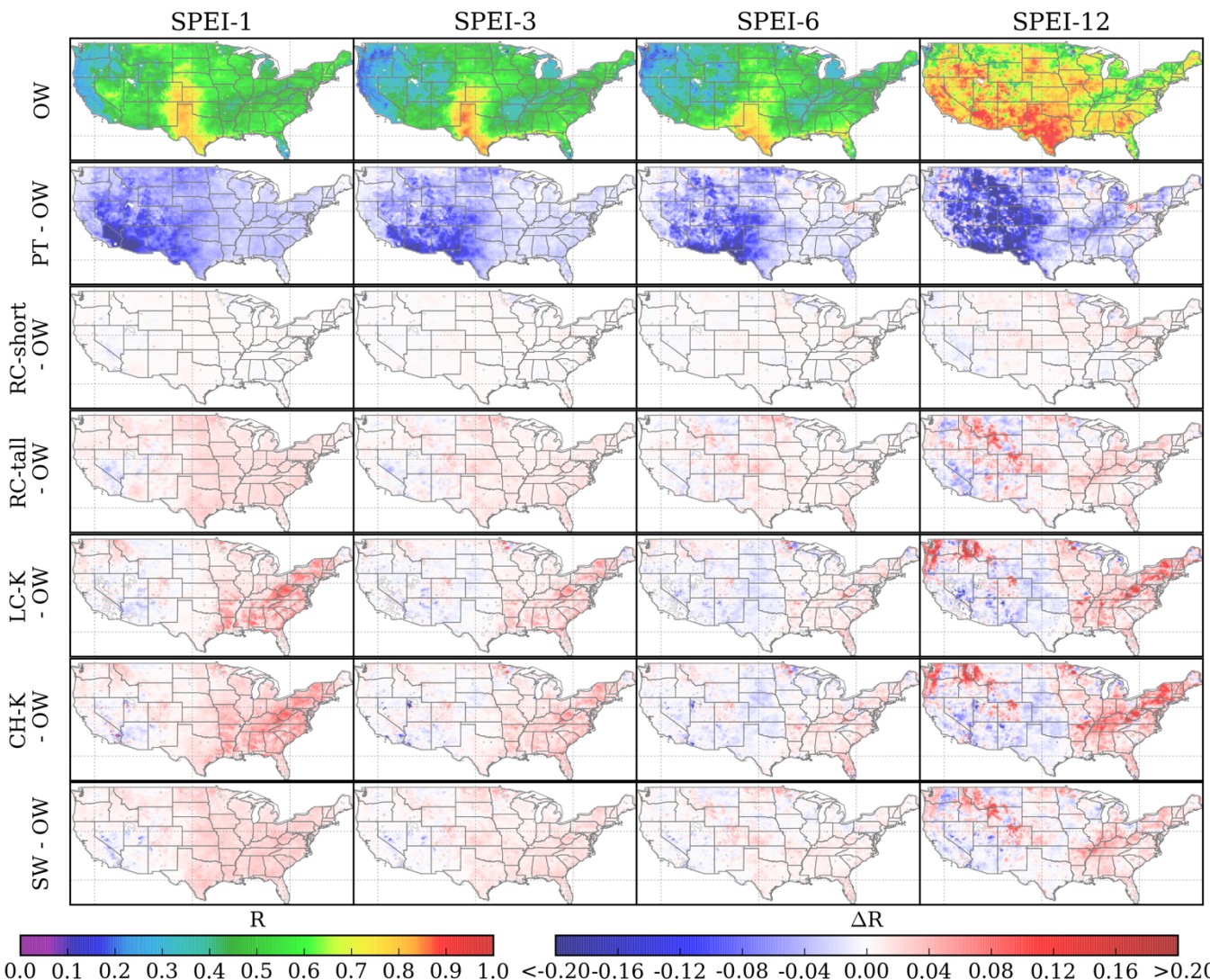

**Figure 6. The first row displays the correlations between SPEI driven by OW and SMsurf. The rows below show the differences in correlations ($\Delta R$) of PET methods relative to OW.**

We further delve into the relative performance of the top four methods summarized by major vegetation types and by aridity (Fig. 7). Both LC and CH increase $R$ significantly in forests, especially in evergreen broadleaf, deciduous broadleaf, and mixed forests, where the largest $\Delta R$ exceeds 0.1, and the average $\Delta R$ hovers around

or above 0.05 for 1-month scale (Fig. 7a). Notable improvements in evergreen needleleaf forest, woody savanna, croplands, and mosaic lands compared to the OW are also observed. LC performs slightly better than CH in forests, while CH performs slightly better in shorter vegetation.

For the time scale of 12-month (Fig. 7c), OW has an already high the average $R$ of 0.73 across the CONUS. LC and CH's performance are outstanding in forests, with an average $\Delta R$ of about 0.05 and the largest $\Delta R$
even exceeding 0.25. In evergreen needleleaf forests, CH and LC's performance are significantly higher than that of the 1-month time scale. In humid regions, CH and LC's improvements over SW becomes even more apparent compared to the 1-month time scale (Fig. 7d).

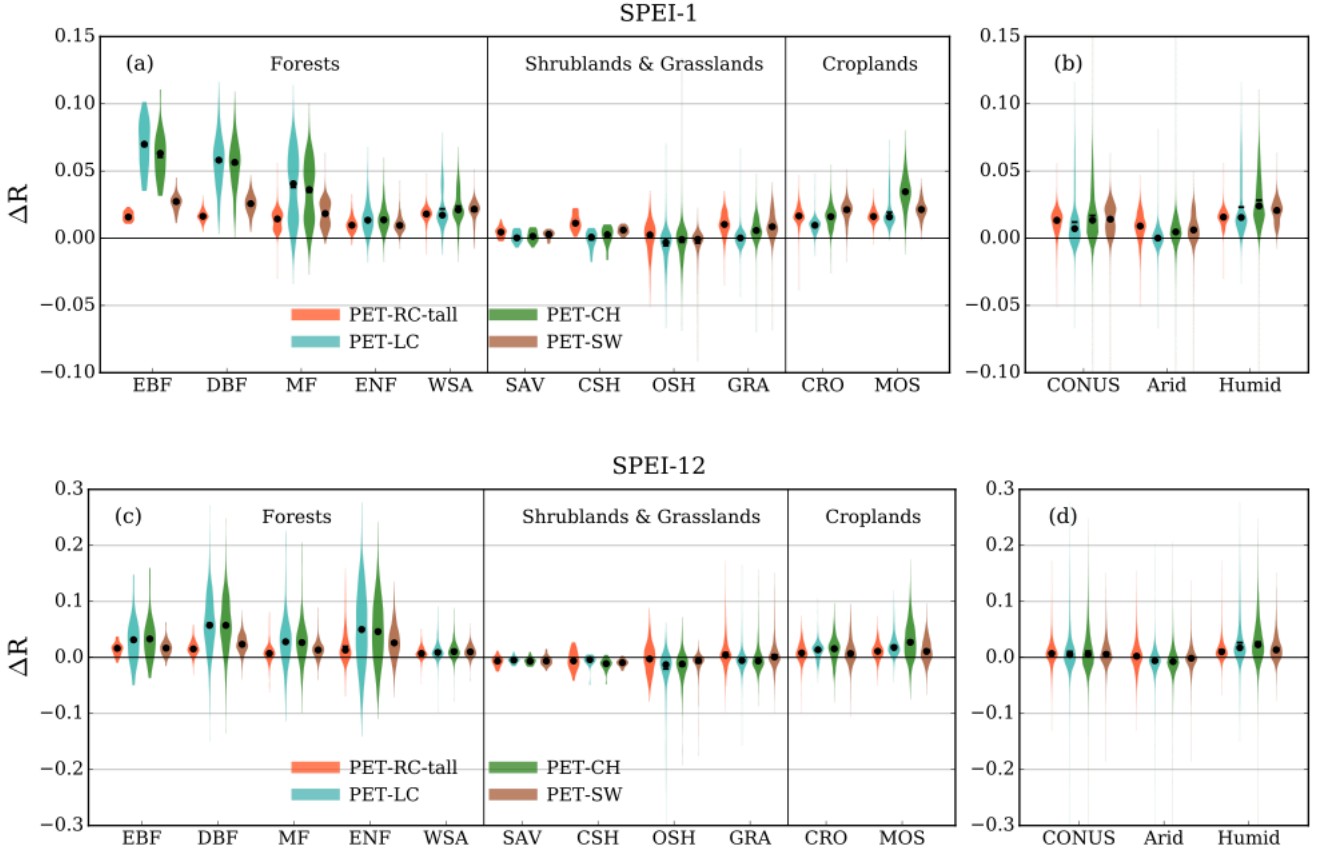

**Figure 7. Violin plots of differences in correlations of three PET methods relative to OW,**
**grouped by vegetation types and aridity. In each violin plot, the black dot represents the median and the black line represents the mean.**

In contrast, the average performance of LC and CH in grasslands, shrublands, and savannas (Fig. 7a and c), which are the dominant vegetation types in the western CONUS, are equivalent to or slightly lower than OW.

The magnitude of averaged $\Delta R$ of LC and CH are slightly smaller than RC-tall and SW, mainly due to their weaker performance in the arid shrublands and grasslands, which cover large portions of the CONUS. The more complicated LC, CH, and SW methods show less advantage or even worse performance than RC-tall and OW in nonforested and arid grid cells (Fig. 7c-d).

## 6 Discussion

### 6.1 Interaction between surface features

Fig. 3 provides important insights into the SPEI sensitivity to different surface features. Introducing $Gs$ with seasonal vegetation dynamics accounts for most of the total improvement of PET algorithm. This confirms that the FAO approaches are more favored than the OW approach due to its constraints on $Gs$. This highlights the importance of leaf area index (LAI) as a vegetation feature for drought depiction. LAI is a scaling factor to upscale $Gst_{max}$ to maximum canopy conductance. This is different from the drought index based on the normalized difference vegetation index (NDVI) or LAI, which requires the real-time dynamics of satellite data. This approach only requires the climatology of LAI, which can be easily implemented for drought forecasting where real-time or near-future data are not available.

Using realistic surface roughness does not necessarily improve the overall performance of the SPEI. In fact, the consistency between aerodynamic conductance and surface conductance is more critical for the skill of PET method. Previous study by Peng et al. (2019) explains the linkage between the ratio of actual ET to PET and the ratio of $Ga$ to $Gs$. When $Ga/Gs$ is large, the ratio of actual ET to PET becomes smaller. Although our study focused on the maximum evapotranspiration given the realistic vegetation condition, such a relationship remains valid. Thus, a large $Ga/Gst_{max}$ ratio should better limit PET with realistic surface constraints. In fact, the LC approach activates surface roughness and increases $Ga$, while constraining $Gst_{max}$ and reducing $Gs$; the CH approach further incorporates canopy height and LAI in the representation of surface roughness. Altogether these factors increase the $Ga/Gs$ ratio and result in significant improvement in capturing the temporal evolution of SMsurf.

### 6.2 Surface characteristics matter in the forests

Our analysis concludes that incorporating surface features can largely improve the accuracy of drought monitoring in the forests. There are two vegetation groups with significantly improved correlation after incorporating the realistic surface characteristics. Forests over the eastern and pacific western U.S., such as evergreen broadleaf and deciduous broadleaf forests, the LC and CH methods exhibit large $\Delta R$ compared to OW (up to 0.12 for 1-month and up to 0.25 for 12-month, Fig. 7a, c). While OW has a $\Delta R$ at about 0.036 compared to the zero PET control scenario (Fig. 4b), LC has an average $\Delta R$ of 0.032 relative to OW in these forests. This means the improvement of LC over control scenario is almost doubling of OW. CH and LC also display a significant increase in $R$ at about 0.025 in woody savanna. The enhancements in the forests or woody savannas are the most predominant since LAI in forests is relatively variable, and surface roughness is also the strongest. Although the southeastern U.S. has a humid subtropic climate, this region also suffered

from periodic droughts in 1986–1988, 1998–2002 and 2006–2009 (Seager, Tzanova, & Nakamura, 2009; Pederson et al., 2012), which is consistent with the increased forest drought severity from 1987-2013 (Peters, Iverson, & Matthews, 2014; Clark et al., 2016). Drought monitoring in these regions is also critical and can benefit from our approach that significantly improve the spatial and temporal accuracy in the forests.

In contrast, the short-grass regions (grasslands, shrublands, and savannas) located in the western U.S. exhibit minimal improvements for LC. The CH method, incorporating the newly available Lang et al. (2023) canopy height dataset, improves the correlations in grasslands, croplands, and mosaic lands. Given that the RC-tall method—a similar big leaf model—performs better than LC and CH in shrublands (Fig. 4, 7), it suggests that uncertainties in LC and CH's $Gst_{max}$ could result in these outcomes. Additionally, a comparison between

$Gst_{max}$ and $Rst_{min}$ (used in SW) highlights uncertainties in this parameter. For instance, $Rst_{min}$ in shrublands, grasslands, and savannas ranges from 100-180 s m$^{-1}$ (equivalent to $Gst_{max}$ of 5-10 mm s$^{-1}$), which is generally lower than 9-12 mm s$^{-1}$ reported by Kelliher et al. (1995). These findings highlight the need for in-situ measurements of surface conductance in these areas.

    Furthermore, these areas have sparse vegetation cover, and thus LAI plays a less effective role in determining

the seasonal dynamics of PET. In the meantime, these areas are located in the arid regions (Fig. 7), the improvements of PET do not have significant effects on modeling the soil moisture, and precipitation dynamics may dominate the soil moisture variations.

**6.3 Strategies for PET method selection**

    Both the CH and LC methods not only provide modest absolute PET values (Fig. 5a, C2) but also display

better performance across many areas (Fig. 6). Specifically, LC and CH estimate an annual PET of roughly 1200 mm, which is within the range of the higher OW value (1424 mm) and the lower values around 1100 mm from RC-short as well as Sun et al. PET dataset (Fig. C2).

    As Ershadi et al. (2015) pointed out, no single model consistently outperformed any other when considered across all land cover types. the selection of PET for model simulation varies depending on the region

(Pimentel et al., 2023). We recommend the use of both LC and CH parameterization for drought monitoring in the forests, in which the roughness and surface conductance parameters vary with realistic vegetation conditions. Both are superior than OW or RC-short because of better performance, and compared to SW, they are both better performing and a simpler approach in the forested areas. Between CH and LC, we recommend CH because it factors in the dynamic change of vegetation structure and provides slightly better

performance in woody savanna.

For shrublands and grasslands, we recommend the use of RC-tall to replace the more widely used RC-short for drought monitoring. We found that the RC-tall approach has a higher skill than the RC-short approach that is more widely used. The main difference between these two methods is the $C_n$ constant that describes the effect of aerodynamic conductance (Allen et al, 2005). The implementation of tall reference ($C_n = 1600$) seems to work better than the short reference ($C_n = 900$) over the CONUS. It is worth noting, however, that the FAO approaches assume a universal $C_n$ regardless of actual vegetation type. The better skill of RC-tall will not always hold, which may overestimate PET in semi-arid non-vegetated regions.

For sparse vegetation, since the responses of the components of evapotranspiration to the environmental drivers are different (Katul, Oren, Manzoni, Higgins, & Parlange, 2012; Or & Lehmann, 2019), the partitioning between canopy and soil can also play a role in determining AED. The SW model significantly improves the SPEI skill driven by the OW approach. It outperforms LC and CH in the croplands and grasslands. Despite its complexity, it is a good choice for drought monitoring in these vegetation types (Sun et al., 2023).

For croplands, we recommend choosing between RC-tall versus RC-short based on the actual crop canopy height. The more realistic approach is to use RC-tall for higher crops. Lastly, the PT method has the poorest correlation with soil moisture and is unlikely to capture drought dynamics.

### 6.4 Bridging gaps in drought prediction

Motivated by the question of whether incorporating surface characteristics can improve drought prediction, we overcome several limitations of previous drought quantification methods. Firstly, our study presents a different approach whereby we focus on the maximum possible evapotranspiration for a given vegetation condition. This concept allows a physically meaningful definition of evaporative demand for the non-uniform land surfaces.

Secondly, the ultimate goal of PET calculation is to simulate ET and to quantify drought. Despite the simplicity of calculating PET using the existing Penman-type methods, the biggest challenge for assessing these methods is validation. Since the real evaporative demand rate is unattainable from observations, it is challenging to validate which PET method is superior directly. Even using ET observations for PET validation can be problematic because biased PET estimates and wrong surface biophysical parameters can still produce accurate ET estimates for locations with ET measurements (Peng et al., 2019). Our study evaluates the PET methods by comparing drought index with independently observed soil moisture (Vicente-Serrano et al., 2012). This approach helps diagnose the most appropriate PET approach for drought quantification directly while avoiding the complexity and divergence caused by various PET definitions.

While the absolute improvements in correlation with soil moisture appear modest, they represent significant percentage changes of average 25-30% and notable local improvements of 86-89% in forests. We acknowledge the need for evaluation of the effectiveness in addition to the temporal correlations. Specifically, future studies should evaluate the capability of the land cover specific approaches to accurately capture extreme events.

Finally, our approach bridges the gap between the two methodologies for quantifying soil-moisture drought, which is of most relevance to agriculture (Seneviratne, 2012). Since soil moisture observations are limited by inadequate measurement networks, drought indices such as the SPEI are often used to quantify drought. In hydrology, a drought index is a simple water balance model driven by surface meteorology without the use of any surface characteristics. Its shortcomings are the neglect of seasonally varying vegetation cover and the incapability to capture the vegetation control on transpiration. An alternative is to use land surface models to estimate large-scale soil moisture (Sheffield, & Wood, 2007). This approach often builds in vegetation dynamics and can provide temporally consistent soil moisture simulations, but it also requires substantial efforts to prepare meteorological forcings at high temporal resolution, set up the domain, spin up, and calibrate. Our approach is a compromise between the above two types of models, which is more realistic and process-based than the commonly used drought index while being easy-to-implement and less data-intensive than a land surface model.


## 7 Conclusions

To understand whether incorporating surface characteristics can improve drought prediction, we revise current PET methods in a newly developed drought index (SPEI), using the concept of maximum ET for any given vegetation condition. We use a simple look-up table approach combining in situ measurements and large-scale data fusion products for the key surface and aerodynamic parameters,. This study also presents a novel application of independent soil moisture observations to diagnose the most appropriate PET approach for drought quantification. Our approach is proved to be more effective than widely used big leaf methods and two source model in accurately predicting soil moisture spatiotemporal dynamics in the forests and humid regions. LAI has a particularly important influence on the skill of the SPEI. This new yet simple approach strikes a balance between a meteorology-driven water balance model and a complex land surface model for drought prediction. It could improve the accuracy of the drought reconstruction in forests and displays great potential to improve real-time drought forecast.

## Appendix A. Shuttleworth-Wallace Model

The Shuttleworth-Wallace (SW) two source model was developed to more accurately represent evapotranspiration from the sparse vegetation. Different from the big leaf models, SW treats the surface as a two-component structure: sparse vegetation (e.g., row crops) and soil. The following formulas are adapted from Equations 11-18 in Shuttleworth and Wallace (1985).

$$PET_{SW} = C_c PET_{PM}^c + C_s PET_{PM}^s \tag{A1}$$

where $PET_{PM}^c$ and $PET_{PM}^s$ are Penman-Monteith like combined equations (Eq. 4) for a closed canopy and
bare soil. Each term is given by the following formulas

$$PET_{PM}^c = \frac{\Delta(R_n - G) + (\rho_a C_p D - \Delta r_a^c (R_n^s - G))/(r_a^a + r_a^c)}{\lambda\left(\Delta + \gamma\left(1 + \frac{r_s^c}{r_a^a + r_a^c}\right)\right)} \tag{A2}$$

$$PET_{PM}^s = \frac{\Delta(R_n - G) + (\rho_a C_p D - \Delta r_a^s (R_n - R_n^s))/(r_a^a + r_a^s)}{\lambda\left(\Delta + \gamma\left(1 + \frac{r_s^s}{r_a^a + r_a^s}\right)\right)} \tag{A3}$$

$$C_c = \frac{1}{1 + \frac{R_c R_a}{R_s(R_c + R_a)}} \tag{A4}$$

$$C_s = \frac{1}{1 + \frac{R_s R_a}{R_c(R_s + R_a)}} \tag{A5}$$

$$R_a = (\Delta + \gamma)r_a^a \tag{A6}$$

$$R_s = (\Delta + \gamma)r_a^s + \gamma r_s^s \tag{A7}$$

$$R_c = (\Delta + \gamma)r_a^c + \gamma r_s^c \tag{A8}$$

where many terms have been given by Eq.1-2, except

$R_n^s$ = net radiation over soil surface = $R_n^s(1 - f_{veg}) = R_n^s \cdot \exp(-0.5 \cdot LAI)$

$r_a^a$ = aerodynamic resistance between canopy height and reference level (s m⁻¹)

$r_s^s$ = surface resistance of the substrate (s m⁻¹)

$r_a^s$ = aerodynamic resistance between substrate and the canopy (s m⁻¹)

$r_s^c$ = bulk stomatal resistance of the canopy (s m⁻¹)

$r_a^c$ = bulk boundary layer resistance of the vegetative elements in the canopy (s m⁻¹).

In this study, the resistances are parameterized for the feasible minimal values based on the water-unlimited assumption for estimating PET. The substrate resistance $r_s^s$ is set to zero s m⁻¹ as a saturated surface. The
canopy resistances are dependent on LAI (Shuttleworth and Wallace, 1985, Equations 19-20).

$$r_s^c = Rst \cdot \frac{1}{LAI_e} \tag{A9}$$

$$r_a^c = r_b \cdot \frac{1}{2LAI} \tag{A10}$$

Stomatal resistance $Rst$ is set to $Rst_{min}$ obtained by the land cover types in Table 1. The effective leaf area index $LAI_e$ is LAI/2 and is capped to 2 (even when LAI is greater than 4). Note that, $r_s^c$ does not have valid values for non-vegetated grid cells (at a specific time of the year or location). The leaf boundary layer resistance $r_b$ is set to a value of 50 s m⁻¹ (Brisson et al., 1998).

The formulas of aerodynamic resistances are given as follows (Shuttleworth and Gurney, 1990; Zhou et al., 2006).

$$r_a^s = \frac{h \cdot \exp(n) \ln\left(\frac{z_m - d_0}{z_0}\right)}{nk^2(h - d_0)} (\exp\left(-\frac{nz_{0g}}{h}\right) - \exp\left(-\frac{n(z_{0m} + d_p)}{h}\right)) \tag{A11}$$

$$r_a^a = \frac{\ln\left(\frac{z_m - d_0}{z_0}\right) \ln\left(\frac{z_m - d_0}{h - d_0}\right)}{k^2 u_z} + \frac{\ln\left(\frac{z_m - d_0}{z_0}\right) h}{nk^2(h - d_0)} (\exp(n\left(1 - \frac{z_{0m} + d_p}{h}\right)) - 1) \tag{A12}$$

where $h$ is canopy height (m), $k$ is the von Karman constant, $z_{0m}$ is the "preferred" roughness length (m), $z_{0m} = h/8$, $d_p$ is the "preferred" zero plane displacement height (m), $d_p = 0.63h$, $z_{0g}$ is the roughness length of ground (m), $u_z$ is the wind speed from the measurement height (m s⁻¹), and $z_m$ is the
measurement height (m), assuming $z_m = h + 2$.

$d_0$ is the zero plane displacement of canopy (m), $n$ is the eddy diffusivity decay constant of the vegetation, and $z_0$ is the canopy roughness length (m). These terms are parameterized as following (Equations 22-26, Zhou et al., 2006):

$$n = \begin{cases} 2.5, & h \leq 1 \\ 2.306 + 0.194h, & 1 < h < 10 \\ 4.25, & h \geq 10 \end{cases} \tag{A13}$$

$$d_0 = \begin{cases} h - z_{0c}/0.3, & LAI \geq 4 \\ 1.1h \cdot \ln(1 + (C_d LAI)^{0.25}), & LAI < 4 \end{cases} \tag{A14}$$

$$z_0 = \min(0.3(h - d_0), z_{0g} + 0.3h(C_d LAI)^{0.5}) \tag{A15}$$

$$C_d = \begin{cases} 1.4 \times 10^{-3}, & h = 0 \\ 0.25\left(-1 + \exp\left(0.909 - \frac{3.03z_{0c}}{h}\right)\right)^4, & h > 0 \end{cases} \tag{A16}$$

$$z_{0c} = \begin{cases} 0.13h, & h \leq 1 \\ 0.139h - 0.009h^2, & 1 < h < 10 \\ 0.05h, & h \geq 10 \end{cases} \tag{A17}$$

where $z_{0c}$ is the roughness length for a closed canopy (m), $C_d$ is the mean drag coefficient for individual
leaves.

## Appendix B. Canopy height data

We evaluated the newly available global canopy height dataset (Lang et al., 2023) and the widely used global tree height dataset (Simard et al., 2011). Although the datasets are highly consistent with each other (Figure B1), some discrepancies exist for low-vegetation cases, which is expected due to the fact that Simard et al. focused on the tree height estimates. We further compared the histogram of canopy height in different land cover types between Lang et al. (Figure B2) and Simard et al. (Figure B3). The two datasets are highly consistent in the forests, while Lang et al. provides valuable information in the short vegetation types.

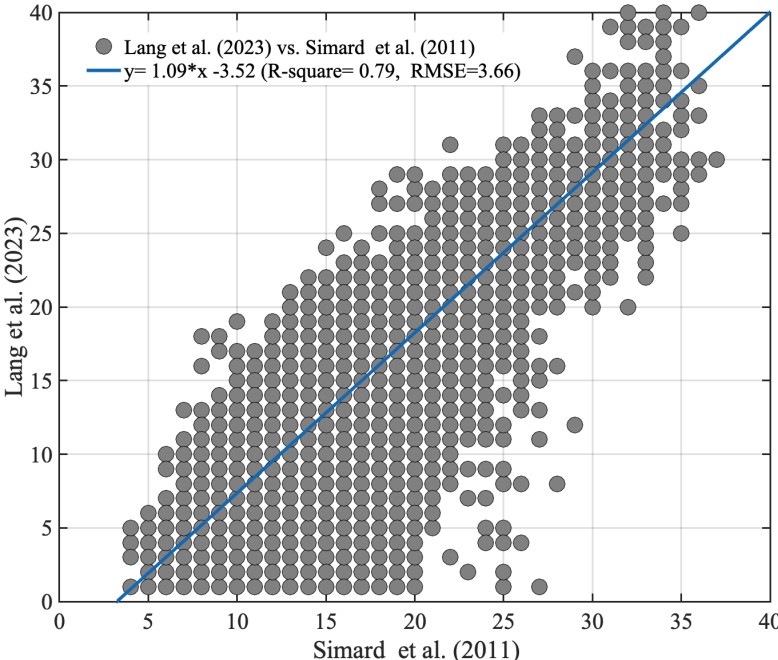

**Figure B1. The comparison in canopy height between Lang et al. (2023) and Simard et al. (2011).**

We reconstruct the canopy height in each grid cell by comparing the value in Lang et al. with the ranges given the land cover type, if it is out of the range (smaller than $h_{min}$ or greater than $h_{max}$) then we give the grid cell a typical value of canopy height ($h_{typ}$). For forests, we continue to follow the definitions from the IGBP land cover classification that the forests are more than 2 m, which we supersede the range in Lang et al. when it gave a range less than 2 m. Typical canopy height is taken from the value of the peak (mode) instead of median for forests. For DBF, Lang et al. only has 3 data points, so we use the distribution of Simard et al. instead, while keeping the lower limit of 6 m from Lang et al.. For grasslands, wetlands, and croplands, the lidar estimates from Lang et al. or Simard et al. are typically more than 3-5 meters, possibly due to the overestimation of the grid cell by the sampling of tall trees. Considering the difficulties in separating trees from the grass and pastures, we did not adopt the high canopy height values in these land cover types. We use conservative estimates from the literature, 1.5 m (mean value of 0-3m) for grasslands and 0.5 m for wetland.

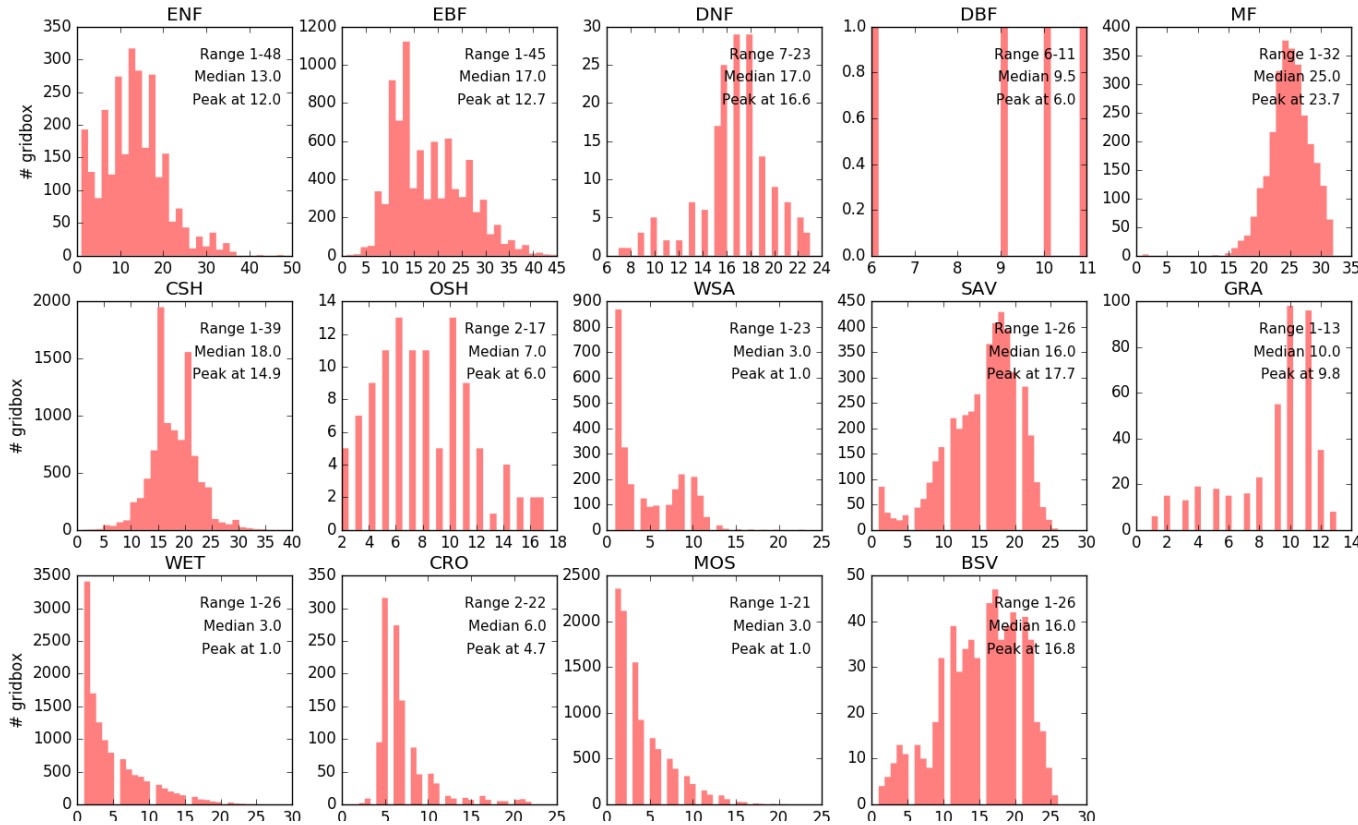

Figure B2. The histogram of canopy height (Lang et al., 2023) by land cover type over the CONUS (excluding non-vegetated land cover: URB, MOS, SNO).

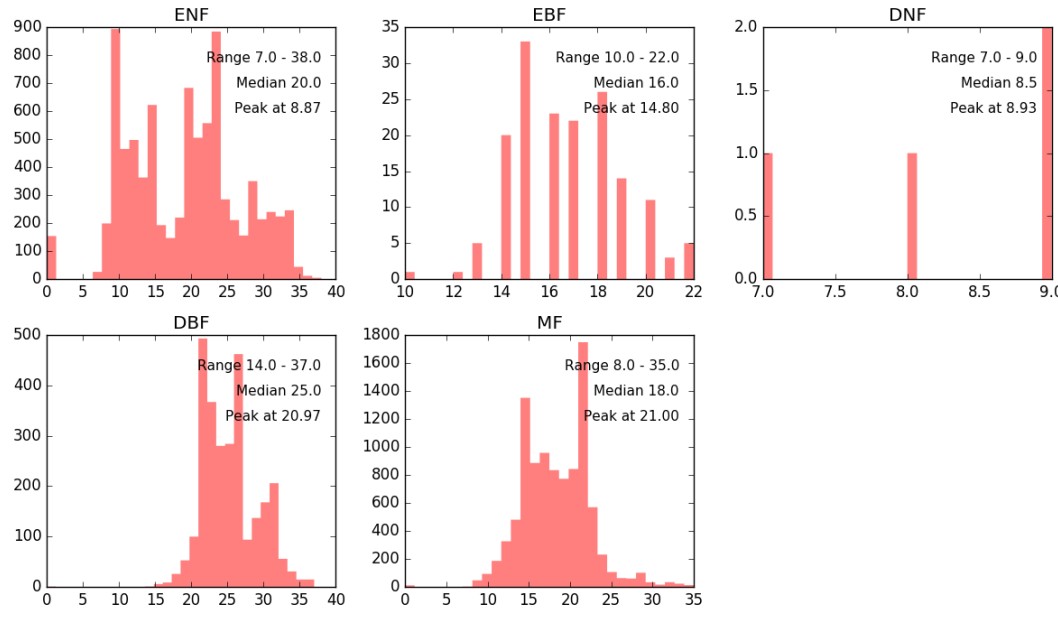

Figure B3. The histogram of tree height (Simard et al., 2023) by forest type over the CONUS.

**Appendix C. Comparison of multiple PET datasets**

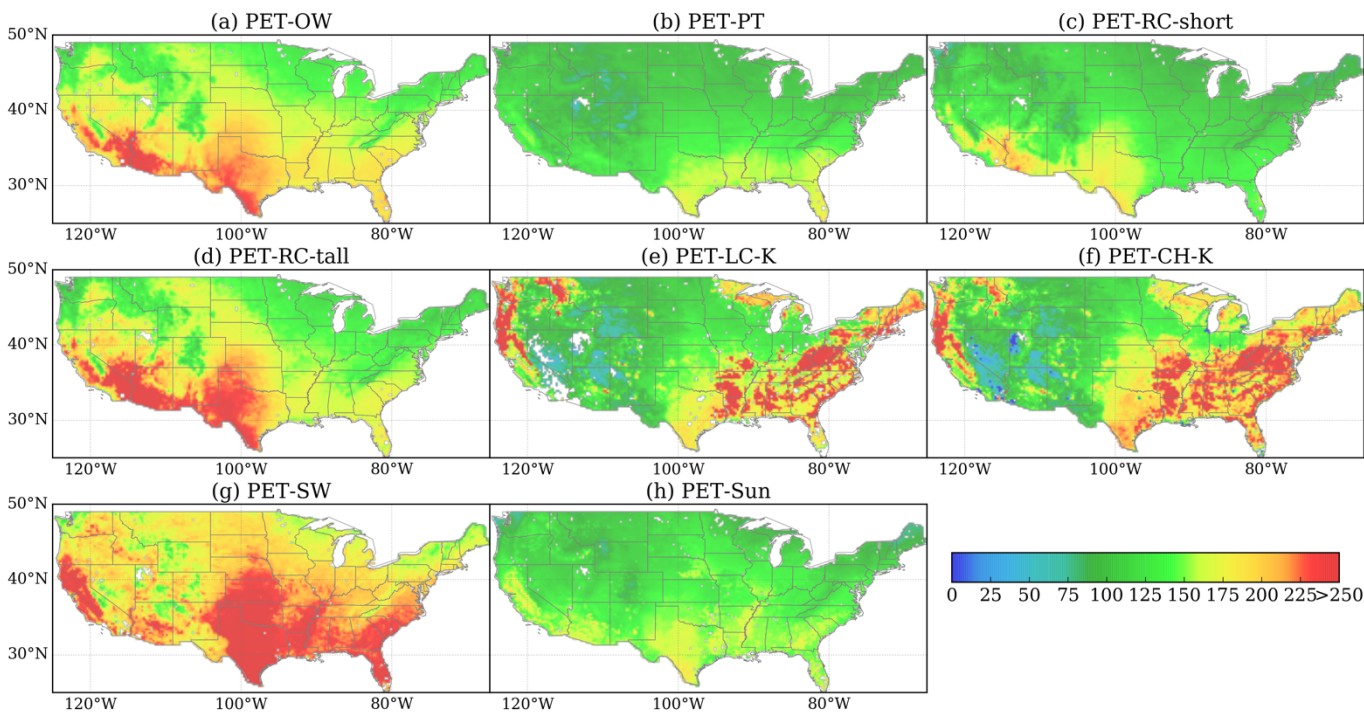

**Figure C1. Growing season averages of the PET methods in this study (Table 4) and the PET dataset by Sun et al. (2023) over the CONUS.**

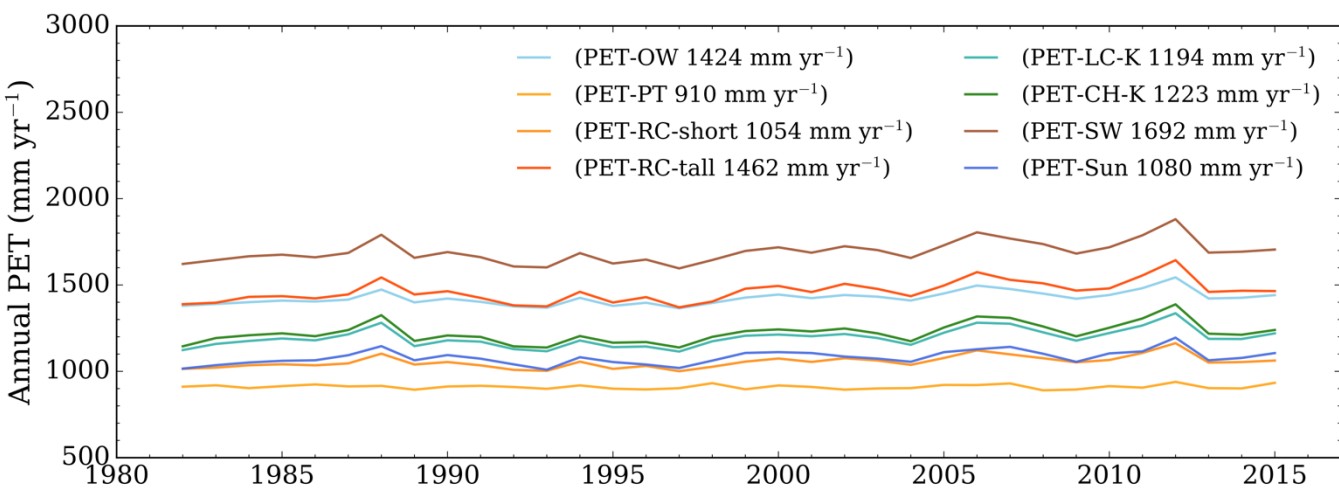

**Figure C2. Annual times series of the PET methods in this study (Table 4) and the PET dataset by Sun et al. (2023) over the CONUS.**

## Code and data availability

The code used to process data and perform analysis for this study is available in the public repository at

https://github.com/pitcheverlasting/spei-pet-evaluation/

The data provided along with this study include the key surface parameters, PET annual data from the main methods, precipitation, and SPEI dataset, available in this public repository:

https://doi.org/10.6084/m9.figshare.12132696.v1.

The primary data and tools can be downloaded from the PRISM Climate Group at Oregon State University (http://www.prism.oregonstate.edu), the ESA CCI soil moisture project team (https://www.esa-soilmoisture-cci.org/node/145), the GIMMS LAI3g product team (https://drive.google.com/open?id=0BwL88nwumpqYaFJmR2poS0d1ZDQ), the Global Land Surface Satellite project (http://www.glass.umd.edu/Download.html), the SPEI R package released by Santiago Beguería and Sergio M. Vicente-Serrano at CSIC in Spain (https://cran.r-project.org/web/packages/SPEI/), the Global Land Cover Climatology project (https://archive.usgs.gov/archive/sites/landcover.usgs.gov/global_climatology.html), and the CDO software (https://code.zmaw.de/projects/cdo).

## Author contributions

LP and JS conceived the idea, LP designed and implemented the PET experiments and analyzed the data, ZW processed some input data, LP wrote the paper with contributions from ZW, ME, and EFW.

## Competing interests

The authors declare that they have no conflict of interest.

## Acknowledgements

This research has been supported by NASA under grant NNX14AB36A. We thank Dan Li at Boston University for the helpful discussions. We thank Philip Lu for proofreading.

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
