# Peer review of "An Enhanced SPEI Drought Monitoring Method Integrating Land"

_EGUsphere, 2023_

## Referee Comment (RC2)

**Review comments**

Peng et al.'s manuscript provides a valuable estimate of global potential evapotranspiration (PET) and forms the basis for developing the SPEI index. The authors incorporate more realistic vegetation characteristics, such as Leaf Area Index (LAI) and conductance, to enhance PET estimation. However, some sections of the manuscript, particularly the structure and descriptions, could benefit from further clarity. The novel aspects of the PET calculation method should be more distinctly highlighted or enhanced.

1. A more detailed description of the "two-source model" in Section 3.3 would be beneficial. The manuscript does not clearly articulate the relationship between this model, Equation (13), and the improved vegetation characteristics described in Section 3.2. The statement "We adopt the same parameterizations detailed in Zhou et al. (2006)" is too vague. It would be valuable to elaborate on how these parameter improvements are integrated into your PET method.

2. The manuscript estimates PET over 1981–2017. This timeframe should be explicitly mentioned in Sections 2 and 3, such as "PET is estimated over 1981–2017 using [specific methodology]."

3. Clarify whether PET calculations are based on monthly or daily scale meteorological inputs. The application of land surface ancillary data in your equations, such as the usage of "black-sky and white-sky albedo," is not clearly explained. For instance, how is albedo factored into the net radiation calculations in your equations?

4. On L121, you mention obtaining "canopy height data from a global tree height dataset at 1-km for 2005 using spaceborne lidar." It seems not clear how this dataset is used in your study? You also state that "As canopy height and frictional velocity are rarely measured continuously for each grid, we use a simple look-up table approach to provide roughness parameters." These statements seem contradictory and need clarification.

5. Section 3.1 lists different PET methods, most of which are derived from the Penman equation. Including the derivation process in the supplementary material and schematic figures illustrating the differences between these methods (e.g., big leaf models vs. twosource models) would enhance understanding. This suggestion is optional if it's difficult to implement.

6. In Section 3.3.3, clarify the role of Gstmax in previous PET methods or equations mentioned earlier.

7. While many surface vegetation characteristics are included to improve PET estimations, some easily accessible characteristics are not utilized. Global canopy vegetation height data (https://www.nature.com/articles/s41559-023-02206-6), which could be employed in Ga estimation, is now available. Other datasets like the 1k datasets (https://essd.copernicus.org/preprints/essd-2023-242/) may also be valuable for your study.

8. A recent study "Sun, S., Bi, Z., Xiao, J., et al." (2023) considers comprehensive parameters for improved PET estimation. If detailed consideration of vegetation characteristics is a novelty of your study, please explicitly explain its advantages compared to this study. Alternatively, if your focus is more on comparing different PET methods with limited vegetation considerations, clarify this in your introduction and discussion.

9. Compare your PET estimations with reference datasets, such as Sun et al. (2023).

10. Appendix A contains important information leading to the results in Section 5.1. Mentioning this in your method sections would prevent sudden introduction of these comparisons in the results. Some sentences around L280 could be moved to the method section.

11. Move Figure A1 to the results section. The results section should feature PET estimations before transitioning to SPEI comparisons (starting in Figure 2).

12. Incorporate multiple soil moisture datasets in your comparison to account for the significant uncertainties among different soil moisture data.

13. On L329, introduce the full name 'LC-Kelliher' before its abbreviation. LC is "land cover" as detailed in the table of Figure 3. Please check the manuscript for any potential similar issues.

14. On L61, provide examples of "conventional PET methods" versus non-conventional methods for clearer understanding. Regarding the statement "The vegetation control on transpiration is often neglected," comment on the impact of plant hydraulics on potential transpiration estimation, referencing relevant studies (e.g., https://agupubs.onlinelibrary.wiley.com/doi/full/10.1029/2018MS001500).

---

## Author Response (AR1)

*Reviewer 1*

**Peng et al. generated a new SPEI drought index by refining the calculation method of potential evapotranspiration (PET), incorporating land surface characteristics driven mainly by leaf area index (LAI). They found that this new SPEI index has a higher correlation with surface moisture data and can explain 29% more variability within soil moisture. The improved index demonstrated good performance in humid regions and forest-dominated ecosystems, making the topic interesting. The manuscript is well-written; however, some concerns regarding methodology and evaluation remain evident. In general, I am favorable to the publication of the manuscript after a thorough revision.**

*Response:* Thank you for your very positive evaluation.

1. **First, it appears that the evaluation throughout the paper relies on the correlation coefficient of the entire time series of SPEI and soil moisture. The increment in the correlation coefficient is almost less than 0.1, even if statistically significant. Since drought indices are typically used to identify and quantify drought events, I suggest the authors evaluate the skill of their improved SPEI index in detecting and quantifying extreme events rather than the dynamics of the entire time series.**

   *Response:* Thank you for the great suggestion. Our study primarily aims to assess the overall improvement in predicting of temporal variations in drought indices and soil moisture. While the absolute increments in correlation appear modest, the percentage change is quite significant, around 25-30%. Despite the small average increase, local improvements are notable (as shown in Figure 4). We acknowledge the above and the importance of capturing the extreme events in discussion section 6.4: "While the absolute improvements in correlation with soil moisture appear modest, they represent significant percentage changes of 25-30% and notable local improvements. We acknowledge the need for evaluation of the effectiveness in addition to the temporal correlations. Specifically, future studies should evaluate the capability of the land cover specific approaches to accurately capture extreme events."

2. **Secondly, I observed that many Ga and Gs parameters (in Table 1) have been used to incorporate features of aerodynamic and surface conductance. I wonder if substantial uncertainty arises from these prescribed parameters. In other words, does the subpar performance of the improved SPEI index in non-forest ecosystems relate to larger uncertainties in parameters for grassland, shrubland, or cropland compared to the forest?**

   *Response:* We agree that the uncertainties in the Gs parameter can potentially affect the results. For example, the better performance of the tall crop reference ET compared to the Land Cover approaches for the non-forest ecosystems suggests inaccuracies in the Gs parameters, given the similar big-leaf model. We compare Gs among the approaches and note the parameter uncertainty in the discussion section 6.2: "Given that the RC-tall method—a similar big leaf model—performs better than LC in these areas (Fig.4), it suggests that uncertainties in LC's $Gst_{max}$ could result in these outcomes. Additionally, a

comparison between $Gst_{max}$ and $Rst_{min}$ (used in SW) highlights uncertainties in this parameter. For instance, $Rst_{min}$ in shrublands, grasslands, and savannas ranges from 100-180 s m$^{-1}$ (equivalent to $Gst_{max}$ of 5-10 mm s$^{-1}$), which is generally lower than 9-12 mm s$^{-1}$ reported by Kelliher et al. (1995). These findings highlight the need for in-situ measurements of surface conductance in these areas."

3. **Thirdly, the improved SPEI exhibits better performance in humid regions, which aligns with expectations given the energy-limited water availability dynamics. However, in arid regions where water availability is more supply-dependent, the adjustment to PET has no significant effects and the uncertainty in precipitation data may be crucial. The authors may elaborate on this point in the manuscript.**

*Response:* Thank you for pointing out the influence of aridity on our results. We acknowledge this in section 6.2: "In the meantime, these areas are located in the arid regions (Fig.7), the improvements of PET do not have significant effects on modeling the soil moisture, and precipitation dynamics may dominate the soil moisture variations".

4. **Specific comment:**
   **Figure 2: It is unclear whether the correlation between soil moisture and SPEI reflects temporal or spatial variability or includes both signals. Additionally, please clarify what the white dots within each bar represent.**

*Response:* The correlation reflects spatially averaged temporal variability between SM and SPEI. The white dots indicate the average difference in correlations between the four methods and the reference method (current Table 2). We will clarify this in the legend of Figure 2: "Differences in spatially averaged correlation (ΔR) of pairs of PET methods that share the same surface characteristics except for one of the surface features: surface roughness, canopy conductance, albedo, and overall consistency among the above features. The white dots indicate the average ΔR between the four methods and the reference method."

*Reviewer 2*

**Peng et al.'s manuscript provides a valuable estimate of global potential evapotranspiration (PET) and forms the basis for developing the SPEI index. The authors incorporate more realistic vegetation characteristics, such as Leaf Area Index (LAI) and conductance, to enhance PET estimation. However, some sections of the manuscript, particularly the structure and descriptions, could benefit from further clarity. The novel aspects of the PET calculation method should be more distinctly highlighted or enhanced.**

*Response:* Thank you for your very positive evaluation.

5. **A more detailed description of the "two-source model" in Section 3.3 would be beneficial. The manuscript does not clearly articulate the relationship between this model, Equation (13), and the improved vegetation characteristics described in Section 3.2. The statement "We adopt the same parameterizations detailed in Zhou et al. (2006)" is too vague. It would be valuable to elaborate on how these parameter improvements are integrated into your PET method.**

   *Response:* Thank you for the suggestion. In order to clarify the parameter improvements for different models, 1) we add a detailed description of the two-source model in Appendix A; 2) we separate the original Section 3.2 into two parts, (a) surface characteristics and (b) parameterizations of surface characteristics, to elaborate how these parameters are integrated into different PET methods.

6. **The manuscript estimates PET over 1981–2017. This timeframe should be explicitly mentioned in Sections 2 and 3, such as "PET is estimated over 1981–2017 using [specific methodology]."**

   *Response:* We add the timeframe in Section 2.1 (P3): "To calculate the SPEI, PET is estimated on daily scale over the period of 1981-2017 using high-quality daily meteorology data from PRISM (Parameter-elevation Regressions on Independent Slopes Model) that employs weather stations and digital elevation model (Daly, Neilson, & Phillips, 1994; Daly et al., 2008).", and in Section 4.1 (P11): "We integrate the PET methods into the SPEI drought index across 1-, 3-, 6-, and 12-month time scales over the CONUS for the period of 1981-2017."

7. **Clarify whether PET calculations are based on monthly or daily scale meteorological inputs. The application of land surface ancillary data in your equations, such as the usage of "black- sky and white-sky albedo," is not clearly explained. For instance, how is albedo factored into the net radiation calculations in your equations?**

   *Response:* PET calculations are based on daily meteorological inputs, which is clarified in Section 2.1 (P3) as mentioned in #6. Regarding the processing of GLASS albedo, we resample the 8-day albedo product to a daily resolution, average the black- and white-sky albedos, and implement gap-filling for missing data using the average of adjacent years.

We add this detail in Section 2.3: "We resample the 8-day albedo to a daily resolution and obtain daily albedo by averaging the black- and white-sky albedos. Missing data are gap-filled using the average of adjacent years."

8. **On L121, you mention obtaining "canopy height data from a global tree height dataset at 1- km for 2005 using spaceborne lidar." It seems not clear how this dataset is used in your study? You also state that "As canopy height and frictional velocity are rarely measured continuously for each grid, we use a simple look-up table approach to provide roughness parameters." These statements seem contradictory and need clarification.**

*Response:* We acknowledge the confusion of using both approaches in this study. We add a new section 3.2.3 Canopy height to clarify the combined usage of global tree height dataset and the literature values for roughness parameters.

"Canopy height ($h$) is a key parameter in determining aerodynamic conductance. The OW and FAO methods generally assume it to be constant across vegetation types and temporal scales. To address this limitation, we introduce two methods for estimating canopy height. The first method, eventually used to obtain $d_0$ and $z_{0m}$ for Eq.9, determines canopy height based on land cover type by calculating the median height within each land cover from the global tree height dataset. The second method, applied in the SW two source model (Appendix A, Eq. A9-10), takes into account both land cover type and dynamic LAI. Each land cover type has a range for canopy height defined by the minimum canopy height ($h_{min}$) and maximum canopy height ($h_{max}$). The actual canopy height is then determined by assuming a linear relationship with LAI following Zhou et al. (2006).

$$h = h_{min} + \frac{(h_{max} - h_{min})LAI}{LAI_{max}} \tag{13}$$

where $LAI_{max}$ represents the annual maximum value at the grid cell level, obtained from the satellite data. Note that h is set to zero if $LAI_{max}$ is zero."

9. **Section 3.1 lists different PET methods, most of which are derived from the Penman equation. Including the derivation process in the supplementary material and schematic figures illustrating the differences between these methods (e.g., big leaf models vs. two-source models) would enhance understanding. This suggestion is optional if it's difficult to implement.**

*Response:* Thank you for the suggestion. We reorganize Section 3.1 by introducing all the existing PET methods at the beginning, including the SW model. We also explain the details of the SW model in Appendix A, as mentioned in #5.

6. **In Section 3.3.3, clarify the role of Gstmax in previous PET methods or equations mentioned earlier.**

*Response:* We clarify the role of Gstmax in previous PET methods in Section 3.2.2: "In previous PET methods, surface conductance is either not considered or assumed to be constant across vegetation types and over time. LAI plays a dominant role in determining the canopy-atmosphere coupling and ET partitioning (Peng et al, 2019; Wei et al., 2017; Forzieri et al., 2020). The OW and PT approach does not consider the role of LAI. The FAO approach uses a constant LAI throughout the growing season. Here we adopt a widely used method in estimating actual ET and assume a well-watered condition."

And "We introduce two options to incorporate an average LAI or the seasonal cycle of LAI into the surface conductance."

7. **While many surface vegetation characteristics are included to improve PET estimations, some easily accessible characteristics are not utilized. Global canopy vegetation height data (https://www.nature.com/articles/s41559-023-02206-6), which could be employed in Ga estimation, is now available. Other datasets like the 1k datasets (https://essd.copernicus.org/preprints/essd-2023-242/) may also be valuable for your study.**

*Response:* Thanks for the recommendations. These datasets are useful for future improvements of our approach. We add this in Section 6.2: "In addition, future improvements to our approach could benefit from incorporating newly available datasets such as Lang et al. (2023) for canopy height."

8. **A recent study "Sun, S., Bi, Z., Xiao, J., et al." (2023) considers comprehensive parameters for improved PET estimation. If detailed consideration of vegetation characteristics is a novelty of your study, please explicitly explain its advantages compared to this study. Alternatively, if your focus is more on comparing different PET methods with limited vegetation considerations, clarify this in your introduction and discussion.**

*Response:* Thanks for pointing out this new study focusing on the Shuttleworth-Wallace model. We clarify the novelty of our study compared to this study in the introduction: "A recent study by Sun et al. (2023) highlighted the importance of incorporating surface properties especially vegetation control in PET and used a two source model designed for sparse vegetation surfaces. However, the model's broader applicability beyond sparse vegetation is uncertain, and additionally it may increase data requirements and associated uncertainties." The advantage of our approach has been illustrated in the end of the discussion: "Our approach is a compromise between the above two types of models, which is more realistic and process-based than the commonly used drought index while being easy-to-implement and less data-intensive than a land surface model."

9. **Compare your PET estimations with reference datasets, such as Sun et al. (2023).**

*Response:* Our estimates align with this study and we note this in the discussion section 6.3: "The LC method not only provides modest absolute PET values (Fig.5a) but also displays better performance across many areas (Fig.6). Specifically, LC estimates an

annual PET of roughly 1200 mm, consistent with PET estimations for the same region as well as temperate zone reported in a recent study (Fig.8 in Sun et al., 2023)."

**10. Appendix A contains important information leading to the results in Section 5.1. Mentioning this in your method sections would prevent sudden introduction of these comparisons in the results. Some sentences around L280 could be moved to the method section.**

*Response:* Thanks for the great suggestion. We move the results and table A1 in original Appendix A to a new section 5.1 Initial assessment of surface characteristics. The original paragraph around L280 has been moved to methods, a new section 4.2, to ensure a smoother transition to the results. Most of the original section 5.1 has been moved to a new section 5.2 for clarity.

**11. Move Figure A1 to the results section. The results section should feature PET estimations before transitioning to SPEI comparisons (starting in Figure 2).**

*Response:* We move the original Appendix A to current results, as addressed in #10.

**12. Incorporate multiple soil moisture datasets in your comparison to account for the significant uncertainties among different soil moisture data.**

*Response:* The ESA CCI SM v4 dataset is chosen for its widely accepted data quality, which is achieved by combining multiple single-sensor active and passive microwave soil moisture products to minimize uncertainty. Gruber et al. (2019) provides a more comprehensive understanding of the data accuracy. We add this in Section 2.2: "The dataset is chosen for its enhanced data reliability by integrating multiple single-sensor active and passive microwave soil moisture products to minimize uncertainty (Gruber et al., 2019)."

**13. On L329, introduce the full name 'LC-Kelliher' before its abbreviation. LC is "land cover" as detailed in the table of Figure 3. Please check the manuscript for any potential similar issues.**

*Response:* We clarify the two land cover (LC) parameterizations for surface conductance in section 3.3: "To calculate surface conductance in Eq. 11-12, we provide two set of parameterizations based on land cover type. The first set is derived from the findings of Kelliher, Leuning, Raupach, & Schulze (1995)… The second set uses the minimum stomatal resistance Rst_min, following Zhou et al. (2006)." We also add a description in a new section 4.3 Comparision of PET parameterizations: "The LC method uses the same aerodynamic conductance method (Eq. 9) but differ in their surface conductance parameterizations: LC-Kelliher, which adopts $Gst_{max}$ from Kelliher, Leuning, Raupach, & Schulze (1995), and LC-Zhou, which uses $Rst_{min}$ from Zhou (2006)."

**14. On L61, provide examples of "conventional PET methods" versus non-conventional methods for clearer understanding. Regarding the statement "The vegetation**

**control on transpiration is often neglected," comment on the impact of plant hydraulics on potential transpiration estimation, referencing relevant studies (e.g., https://agupubs.onlinelibrary.wiley.com/doi/full/10.1029/2018MS001500).**

*Response:* In the line mentioned, we differentiate between "conventional" PET methods, which often assume no or simple universal vegetation control on transpiration, and "non-conventional" methods that account for vegetation control based on specific conditions. For instance, conventional methods that are often used in SPEI include the Thornthwaite and Hargreaves-Samani equations, as well as the Penman open water or Reference crop ET formulas. The "unconventional" methods in this study do not refer to the land surface models or dynamic vegetation models, which normally have representations of the transpiration process including plant hydraulics. This is clarified in L61.

---

## Referee Report (RR1)

**Review comments**

Peng et al. have responded to the reviewers' comments, but there are significant concerns that remain inadequately addressed. Therefore, I recommend a MAJOR revision.

1. The novelty of this study lies in the PET computation, as the authors state, "This study proposes to incorporate surface vegetation characteristics, such as vegetation dynamics data, aerodynamic, and physiological parameters, into existing potential evapotranspiration (PET) methods."

   The authors highlight the inclusion of surface vegetation characteristics—such as vegetation dynamics data, aerodynamic, and physiological parameters—into existing potential evapotranspiration (PET) methods as a novel aspect of their study. However, despite recognizing the importance of these characteristics, the study does not fully utilize the most recent datasets available. Notably, new datasets like the global canopy height dataset released approximately 2 years ago and the global 1k datasets mentioned in prior reviews offer valuable insights. The paper by Sun et al. (2023), while utilizing a different model, aims to leverage the most current global datasets possible. To align with current scientific advancements and fulfill the novelty criteria of the ESSD journal, it is imperative that the authors consider incorporating these more recent datasets into their analysis.

2. Regarding the modification, "A recent study by Sun et al. (2023) highlighted the importance of incorporating surface properties, especially vegetation control, in PET and used a two-source model designed for sparse vegetation surfaces. However, its applicability beyond sparse vegetation remains unclear, raising questions about data requirements and potential uncertainties." It's unclear why the S-W model used by Sun et al. (2023) is deemed unsuitable for areas beyond sparse vegetation, without further explanation or references.

3. A main focus here is on the PET products, and there are numerous available PET datasets over CONUS. However, the comparison between your products and other reference datasets is lacking. As noted in previous comments, the authors only provide a visual comparison, "The LC method not only yields modest absolute PET values (Fig.5a) but also demonstrates better performance across many regions (Fig.6). Specifically, LC estimates an annual PET of approximately 1200 mm, aligning with PET estimates for the same region and temperate zones

reported in a recent study (Fig.8 in Sun et al., 2023)." A more comprehensive comparison with other reference PET datasets seems necessary. It appears that the response from the authors does not attempt to resolve the issues but rather tries to avoid directly addressing the comments.

4. Attempting to access the data, I found it currently unavailable. Does ESSD typically require data accessibility for reviewers? Additionally, statements like "The data generated in our study are published in this public repository: https://doi.org/10.6084/m9.figshare.12132696.v1 (active after acceptance)" are too vague. Specific details regarding data accessibility (e.g., which specifical datasets) should be provided.

---

## Author Response (AR2)

*Reviewer 1*

**Peng et al. have devoted considerable efforts to improve their analysis. This revision successfully addresses the major concerns I had.**

*Response:* Thank you very much for your very positive evaluation.

*Reviewer 2*

**Peng et al. have responded to the reviewers' comments, but there are significant concerns that remain inadequately addressed. Therefore, I recommend a MAJOR revision.**

*Response:* Thank you for your constructive feedback.

1. **The novelty of this study lies in the PET computation, as the authors state, "This study proposes to incorporate surface vegetation characteristics, such as vegetation dynamics data, aerodynamic, and physiological parameters, into existing potential evapotranspiration (PET) methods."**

   **The authors highlight the inclusion of surface vegetation characteristics—such as vegetation dynamics data, aerodynamic, and physiological parameters—into existing potential evapotranspiration (PET) methods as a novel aspect of their study. However, despite recognizing the importance of these characteristics, the study does not fully utilize the most recent datasets available. Notably, new datasets like the global canopy height dataset released approximately 2 years ago and the global 1k datasets mentioned in prior reviews offer valuable insights. The paper by Sun et al. (2023), while utilizing a different model, aims to leverage the most current global datasets possible. To align with current scientific advancements and fulfill the novelty criteria of the ESSD journal, it is imperative that the authors consider incorporating these more recent datasets into their analysis.**

*Response:* Thanks for emphasizing the importance of utilizing the most recent datasets to enhance the novelty of our study. To advance our usage of vegetation characteristics data, we have incorporated the global canopy height dataset, mentioned in prior reviews and in the paper by Sun et al. (2023), into our analysis.

Regarding the global canopy height dataset, we have carefully reviewed the dataset and assessed its potential to improve our analysis. We first compared the newly developed 10m dataset (Lang et al., 2023) with Simard et al. (2011), which was originally used in our study. We remapped 10-m source points to target cells (0.125 degrees) by calculating their mean, with each target cell containing the mean value from all source points within it. Although the datasets are highly consistent with each other (Figure R1), some discrepancies exist for low-vegetation cases, which is expected due to the fact that Simard et al. focused on the tree height estimates.

[Figure]

**Figure R1**. The comparison in canopy height between Lang et al. and Simard et al.

We then further compared the histogram of canopy height in different land cover types between Simard et al. (Figure R2) and Lang et al. (Figure R3). We confirm that the two datasets are highly consistent in the forests, while the Lang et al. provides valuable information in the short vegetation types and indeed could be utilized in our study.

Based on the distribution of canopy height in Lang et al. data for CONUS (0.125 degrees), we update the prior tree height ranges and typical values and expand to other vegetated land cover types (Table R1). We reconstruct the canopy height in each grid cell by comparing the value in Lang et al. with the ranges given the land cover type, if it is out of the range (smaller than $h_{min}$ or greater than $h_{max}$) then we give the grid cell a typical value of canopy height ($h_{typ}$). For forests, we continue to follow the definitions from the IGBP land cover classification that the forests are more than 2 m, which we supersede the range in Lang et al. when it gave a range less than 2 m. Typical canopy height is taken from the value of the peak (mode) instead of median for forests. For DBF, Lang et al. only has 3 data points, so we use the distribution of Simard et al. instead, while keeping the lower limit of 6 m from Lang et al.. For grasslands, wetlands, and croplands, the lidar estimates from Lang et al. or Simard et al. are typically more than 3-5 meters, possibly due to the overestimation of the grid cell by the sampling of tall trees. We used conservative estimates (1.5 m for grasslands and 0.5 m for wetlands) from the literature and did not adopt the high canopy height values in these land cover types.

[Figure]

**Figure R2**. The histogram of tree height of Simard et al. for different forest types over the CONUS.

[Figure]

**Figure R3**. The histogram of canopy height of Lang et al. for different vegetated land cover types over the CONUS.

**Table R1. Canopy height parameters by IGBP land cover\* under previous and revised versions.**

| ID | Code | Name | Previous | | | Revised | | |
|---|---|---|---|---|---|---|---|---|
| | | | $h_{min}$ (m) | $h_{max}$ (m) | $h_{typ}$ (m) | $h_{min}$ (m) | $h_{max}$ (m) | $h_{typ}$ (m) |
| 0 | WB | Water body | 0.001 | 0.02 | 0.01 | 0.001 | 0.02 | 0.01 |
| 1 | ENF | Evergreen needleleaf | 2 | 50 | 18 | 2 [a] | 48 [b] | 13 [b] |
| 2 | EBF | Evergreen broadleaf | 2 | 50 | 30 | 2 [a] | 45 [b] | 17 [b] |
| 3 | DNF | Deciduous needleleaf | 2 | 50 | 15 | 7 [b] | 23 [b] | 17 [b] |
| 4 | DBF | Deciduous broadleaf | 2 | 50 | 17 | 6 [b] | 37 [c] | 9.5 [c] |
| 5 | MF | Mixed forest | 2 | 50 | 19 | 2 [a] | 32 [b] | 25 [b] |
| 6 | CSH | Closed shrublands | 0.1 | 5 | 4 | 1 [b] | 39 [b] | 14.9 [b] |
| 7 | OSH | Open shrublands | 0.1 | 5 | 2 | 2 [b] | 17 [b] | 6 [b] |
| 8 | WSA | Woody savannas | 2 | 30 | 14 | 1 [b] | 23 [b] | 1 [b] |
| 9 | SAV | Savannas | 2 | 30 | 8 | 1 [b] | 26 [b] | 17.7 [b] |
| 10 | GRA | Grasslands | 0.1 | 3 | 0.5 | 0.1 | 3 | 1.5 |
| 11 | WET | Permanent wetlands | 0.1 | 5 | 0.5 | 0.1 | 5 | 0.5 |
| 12 | CRO | Croplands | 0.1 | 5 | 1 | 0.1 | 5 | 1 |
| 13 | URB | Urban and built up | 2 | 50 | 13 | 2 | 50 | 13 |
| 14 | MOS | Cropland/vegetation | 0.1 | 30 | 12 | 0.1 | 21 | 12 |
| 15 | SNO | Snow/ice | 0.001 | 0.02 | 0.01 | 0.001 | 0.02 | 0.01 |
| 16 | BSV | Barren | 0.01 | 0.1 | 0.05 | 0.01 | 0.1 | 0.05 |

\*The above estimates are collected from [a]IGBP classification, [b]Lang et al. (2023), [c]Simard et al. (2023), otherwise based on authors' best estimates.

[Figure]

**Figure R4**. Left: The reconstructed canopy height based on Lang et al. and different data sources including Simard et al.; Right: The canopy height based on land cover types and Simard et al. for forests over the CONUS.

Previously we used the spatial map from Simard et al. derived from lidar data only for the forests (ENF, EBF, DNF, DBF, MF), and only used literature values for short vegetation. The reconstructed canopy height (Fig. R4 left panel) has much more spatial details, especially for non-forest grid cells, than the previous approach mostly based on land cover (Fig. R4 right panel). We admitted that this is indeed one of the limitations in our previous data source and did not clarify our processing and selection. We use the new data from Lang et al. and constrain the range using the distribution within the CONUS.

In addition to this new dataset, we add a new Canopy Height based PET approach (CH) to incorporate the dynamic canopy height dataset, rerun our models, and update our findings accordingly. We clarify the updated approach in the Data and Methods sections as below:

1) We updated the data description in L135: "This study uses the newly developed 10-m global canopy height dataset that merges the Global Ecosystem Dynamics Investigation (GEDI) space-borne LiDAR height data with Sentinel-2 satellite data (Lang et al., 2023). The original 10-m resolution was remapped to 0.125o using the average. Additionally, this study uses a global tree height dataset at 1-km for 2005 using spaceborne lidar (Simard et al., 2011) for complementary analysis in the forests (Appendix B)."

2) We added the CH approach for aerodynamic conductance in L225-240.

3) We documented in detail how we use the canopy height information in the two approaches, LC and CH, in L275: "The first method uses literature values and is adopted in the Land Cover approach (LC, Eq.9). For most of the land cover types (ID 6-16), we applied the values from the look up table except for the forests, where we determined canopy height by calculating the median height within each land cover from the tree height lidar data (Simard et al., 2011).

   The second more comprehensive method is adopted in the Canopy Height approach (CH, Eq. 11) and the SW two source model (Appendix A, Eq. A9-10). It takes into account three factors: land cover type, measured canopy height, and dynamic LAI. We overlayed the land cover map (Fig. 1) and the canopy/tree height data (Lang et al., 2023; Simard et al., 2011) to obtain the distribution in each land cover type (Appendix B). Based on the distribution of the two datasets, land cover definition, and literature ranges, we estimated the minimum canopy height (h_min) and maximum canopy height (h_max) by land cover type (Table 2). As for quality control, we set the outlier (smaller than h_min or greater than h_max) to a typical value of canopy height given land cover type (h_typ, obtained through the mode of the distribution). The actual canopy height is then determined by assuming a linear relationship with dynamic LAI following Zhou et al. (2006)."

4) We added a Table 2 for the canopy height parameters for each land cover type.
5) We added a Table 4 for the detailed formula and parameterization for each PET method.
6) We added an Appendix B to describe the evaluation and processing of canopy height.

With the updated CH approach, we re-ran our models, updated the Figures 4-7 by adding the new CH approach, and rewrote the results and findings in sections 6.1-6.3 surrounding these figures.

Figure R5 (Fig. 4 in the manuscript) shows a significant improvement in the CH method relative to the LC method in non-forested areas (Fig. R5c), highlighting the uncertainty of these parameters in the sparse vegetation and the importance of incorporating actual canopy height. We believe the newly implemented dataset will significantly strengthen the findings of our study. Thank you for your suggestion.

[Figure]

**Figure R5**. Differences in correlations (**ΔR**) for selected PET methods versus the control scenario (PET = 0). Correlations were computed between the 1-month SPEI and SMsurf series across: (a) CONUS, (b) forested grids, and (c) nonforested grids.

2. **Regarding the modification, "A recent study by Sun et al. (2023) highlighted the importance of incorporating surface properties, especially vegetation control, in PET and used a two- source model designed for sparse vegetation surfaces. However, its applicability beyond sparse vegetation remains unclear, raising questions about data requirements and potential uncertainties." It's unclear why the S-W model used by Sun et al. (2023) is deemed unsuitable for areas beyond sparse vegetation, without further explanation or references.**

*Response:* Thanks for pointing out the lack of references that discuss the application of the S-W model across different vegetation types in this paragraph. We added the supporting literature and modified the original sentence in L75 as: "A recent study by Sun et al. (2023) highlighted the importance of incorporating surface properties especially vegetation control in PET and used a two source Shuttleworth-Wallace (SW) model designed and validated for sparse and fragmented vegetation surfaces. However, without further calibration and parameterization, the SW model's broader applicability beyond sparse vegetation is uncertain, and additionally it may increase data requirements and associated uncertainties (Gao et al., 2021; Abeysiriwardana et al., 2022)".

3. **A main focus here is on the PET products, and there are numerous available PET datasets over CONUS. However, the comparison between your products and other reference datasets is lacking. As noted in previous comments, the authors only provide a visual comparison, "The LC method not only yields modest absolute PET values (Fig.5a) but also demonstrates better performance across many regions (Fig.6). Specifically, LC estimates an annual PET of approximately 1200 mm, aligning with PET estimates for the same region and temperate zones reported in a recent study (Fig.8 in Sun et al., 2023)." A more comprehensive comparison with other reference PET datasets seems necessary. It appears that the response from the authors does not attempt to resolve the issues but rather tries to avoid directly addressing the comments.**

*Response:* Thanks for suggesting a more comprehensive comparison between our PET products and other reference datasets. To strengthen our analysis, we have downloaded their datasets, processed to the same spatial resolution, and compared our LC and CH methods with the proposed PET dataset by Sun et al. (2023). The comparison shows that significant spatial pattern differences can be found (Fig. R6), although the time series variations are highly consistent (Fig. R7). The magnitude of Sun et al. is closer to the PT and RC-short methods. The LC and CH methods have medium magnitudes among all the PET methods.

[Figure]

**Figure R6.** Growing season averages of the PET methods in this study (Table 4) and the PET dataset by Sun et al. (2023) over the CONUS.

[Figure]

**Figure R7.** Annual times series of the PET methods in this study (Table 4) and the PET dataset by Sun et al. (2023) over the CONUS.

In the revised manuscript, we added Appendix C with the two figures. We also discuss them in section 5.3 Spatial patterns analysis as well as section 6.3: "Both the CH and LC methods not only provide modest absolute PET values (Fig. 5a, C2) but also display better performance across many areas (Fig. 6). Specifically, LC and CH estimate an annual PET of roughly 1200 mm, which is within the range of the higher OW value (1424 mm) and the lower values around 1100 mm from RC-short as well as Sun et al. PET dataset (Fig. C2). As Ershadi et al. (2015) pointed out, no single model consistently outperformed any other when considered across all land cover types. the selection of PET for model simulation varies depending on the region (Pimentel et al., 2023)". We believe that incorporating the most recent and relevant global datasets can strengthen the novelty and scientific contribution of our study.

4. **Attempting to access the data, I found it currently unavailable. Does ESSD typically require data accessibility for reviewers? Additionally, statements like "The data generated in our study are published in this public repository: https://doi.org/10.6084/m9.figshare.12132696.v1 (active after acceptance)" are too vague. Specific details regarding data accessibility (e.g., which specific datasets) should be provided.**

*Response:* Thanks for pointing out the issues with the need for clearer information regarding the availability of our datasets. The previous provided link was intended to be activated post-acceptance. We will ensure it active and meet the standards of the journal. Here is the private link for reviewers to access the data: https://figshare.com/s/6e1eb7c9f2b6fd13edba

We provide the inputs including key parameters like canopy height and roughness, and outputs and add the following in the data statement so that it is clearly described in the manuscript: "The data provided along with this study include the key surface parameters, PET annual data from the

main methods, precipitation, and SPEI dataset, available in this public repository: https://doi.org/10.6084/m9.figshare.12132696.v1".

To further enhance our transparency, we also add a code statement in the manuscript: "The code used to process data and perform analysis for this study is available in the public repository at https://github.com/pitcheverlasting/spei-pet-evaluation/"

In addition to all the above updates, we also proofread the manuscript, updated the figures and tables (like Table 3 and Fig. 2), corrected any remaining typos, resolved any conflicts/confusion in the previous draft.

**References:**

Abeysiriwardana, H. D., Muttil, N., & Rathnayake, U. (2022). A comparative study of potential evapotranspiration estimation by three methods with FAO Penman–Monteith method across Sri Lanka. *Hydrology*, *9*(11), 206.

Ershadi, A., McCabe, M. F., Evans, J. P., & Wood, E. F. (2015). Impact of model structure and parameterization on Penman–Monteith type evaporation models. *Journal of Hydrology*, *525*, 521-535.

Gao, G., Feng, Q., Liu, X., & Zhao, Y. (2021). Measuring and modeling evapotranspiration of a Populus euphratica forest in northwestern China. *Journal of Forestry Research*, *32*(5), 1963-1977.

Pimentel, R., Arheimer, B., Crochemore, L., Andersson, J. C. M., Pechlivanidis, I. G., & Gustafsson, D. (2023). Which potential evapotranspiration formula to use in hydrological modeling world-wide? *Water Resources Research*, 59, e2022WR033447. https://doi.org/10.1029/2022WR033447